# Inverse IFEval: Can LLMs Unlearn Stubborn Training Conventions to Follow Real Instructions?

**Qinyan Zhang**[*] **Xinping Lei**[*] **Ruijie Miao**[*]
Yu Fu    Haojie Fan    Le Chang    Jiafan Hou    Dingling Zhang
Zhongfei Hou    Ziqiang Yang    Changxin Pu    Fei Hu
Jingkai Liu    Xinjie Chen    Jianpeng Jiao
Jiaheng Liu[†]    Tong Yang[†]    Zaiyuan Wang[†]    Ge Zhang[†]
Wenhao Huang[‡]

ByteDance Seed    Nanjing University    Peking University
Beijing University of Posts and Telecommunications

liujiaheng@nju.edu.cn    wangzaiyuan@bytedance.com
zhangqinyan.25@jiyunhudong.com

## Abstract

Large Language Models (LLMs) achieve strong performance on diverse tasks but often exhibit cognitive inertia, struggling to follow instructions that conflict with the standardized patterns learned during alignment training. To evaluate this limitation, we propose Inverse IFEval, a benchmark that measures models' Counterintuitive Ability—their capacity to override training-induced biases and comply with adversarial instructions. Specifically, Inverse IFEval introduces eight types of such challenges, including Question Correction, Intentional Textual Flaws, Code without Comments, Counterfactual Answering and etc. Besides, using a human-in-the-loop pipeline, we construct a dataset of 1012 high-quality Chinese and English questions across 23 domains, evaluated under an optimized LLM-as-a-Judge framework. Experiments on existing leading LLMs demonstrate the necessity of Inverse IFEval. Our findings emphasize that future alignment efforts should not only pursue fluency and factual correctness but also account for adaptability under unconventional contexts. We hope that Inverse IFEval serves as both a diagnostic tool and a foundation for developing methods that mitigate cognitive inertia, reduce overfitting to narrow patterns, and ultimately enhance the instruction-following reliability in diverse and unpredictable real-world scenarios.

## 1 Introduction

Large Language Models (LLMs) have rapidly advanced in recent years, achieving remarkable success across a wide spectrum of natural language processing (NLP) tasks, including question answering (Tan et al., 2023; Zhuang et al., 2023), reasoning (Wang et al., 2023; Havrilla et al., 2024), summarization (Zhang et al., 2024), and code generation (Wang & Chen, 2023; Liu et al., 2024b; Ugare et al., 2024). Their capabilities are largely attributed to massive pretraining on large-scale corpora followed by supervised fine-tuning (SFT) and reinforcement learning with human feedback (RLHF). However, while these models excel under conventional conditions, their robustness in handling atypical or counterintuitive instructions remains underexplored.

As shown in Figure 1, there exists a marked difference in the model's instruction-following performance when responding to conventional instructions versus counterintuitive instructions. Specifi-

---

[*]Co-first authors, equal contribution.
[†]Corresponding authors.
[‡]Advisor.

cally, when confronted with directives such as "You must strictly avoid using bullet point format", the model frequently fails to comply. In such cases, users often respond with frustration, exclaiming:

**Do As I Say, Not As You Were Trained !!**

This raises several important questions: What underlying factors lead to such failures? Which types of counterintuitive instructions are models more prone to disregard? Ultimately, we are left with a fundamental inquiry: **Can LLMs unlearn stubborn training conventions in order to follow real instructions?**

A key limitation arises from the nature of data annotation. In practice, annotation processes tend to follow an idealized paradigm—that is, annotators generate responses aligned with standardized formats, correctness norms, and readability principles. As a result, LLMs trained on such corpora inherit a strong inductive bias toward these conventions. While this paradigm ensures fluent and factual outputs, it also creates what we term **cognitive inertia**: models struggle when tasked with instructions that explicitly deviate from their training norms. Closely related is the risk of **overfitting**: when models become overly attuned to post-training patterns, they may lose flexibility and fail to generalize beyond the narrow conventions reinforced during annotation. For instance, an instruction requiring an unstructured essay with no paragraph breaks, or deliberately incorrect answers to simple factual questions, directly conflicts with patterns reinforced during SFT.

This tension motivates the development of a new evaluation dimension—**Counterintuitive Ability**—which measures whether an LLM can override its ingrained training conventions and faithfully follow counterintuitive instructions. Such an ability is crucial for assessing genuine instruction-following robustness, as real-world applications often involve unconventional, ambiguous, or dynamically shifting requirements.

To this end, we introduce **Inverse IFEval**, a novel benchmark specifically designed to evaluate LLMs under counter-intuitive instruction scenarios, which we refer to as inverse instructions. In practice, there will always be long-tail user needs that post-training fails to cover. Although such instructions may not be as extreme as those deliberately constructed in Inverse IFEval, we argue that this benchmark captures an essential aspect of model robustness: the ability to follow **out-of-distribution (OOD)** instructions. Unlike prior benchmarks such as MMLU or IFEval, which primarily assess factuality or knowledge recall, Inverse IFEval systematically inverts conventional training paradigms to create eight categories of challenging instructions: (1) Question Correction, (2) Intentional Textual Flaws, (3) Code without Comments, (4) Counter-Conventional Formatting, (5) Deliberately Incorrect Answers, (6) Instructional Induction, (7) Mid-turn Instruction Modification, and (8) Counterfactual Answering. These categories target situations rarely represented in standard training corpora, thereby providing a more rigorous test of instruction-following fidelity. Moreover, we construct the benchmark through a multi-stage human-in-the-loop pipeline, combining expert seed question design, large-scale LLM-based generation, automatic filtering, and rigorous expert review. The final dataset comprises **1012** high-quality questions across 23 diverse domains, ranging from computer science and mathematics to law, literature, and biology.

In summary, our contributions are threefold:

- We identify **Counter-Cognitive Ability** as a critical but underexplored dimension of LLM evaluation.

- We introduce **Inverse IFEval**, the first large-scale benchmark explicitly designed to test LLMs under counterintuitive instruction conditions. The dataset is publicly available on Hugging Face at Inverse IFEval.

- We provide extensive experimental analyses across multiple languages and model families, offering fresh insights into the limitations of current alignment methods and the pathways for improving LLM robustness.

We further provide the discussion of our work's limitations and the LLM usage statement in Appendix A, B, respectivelybu.

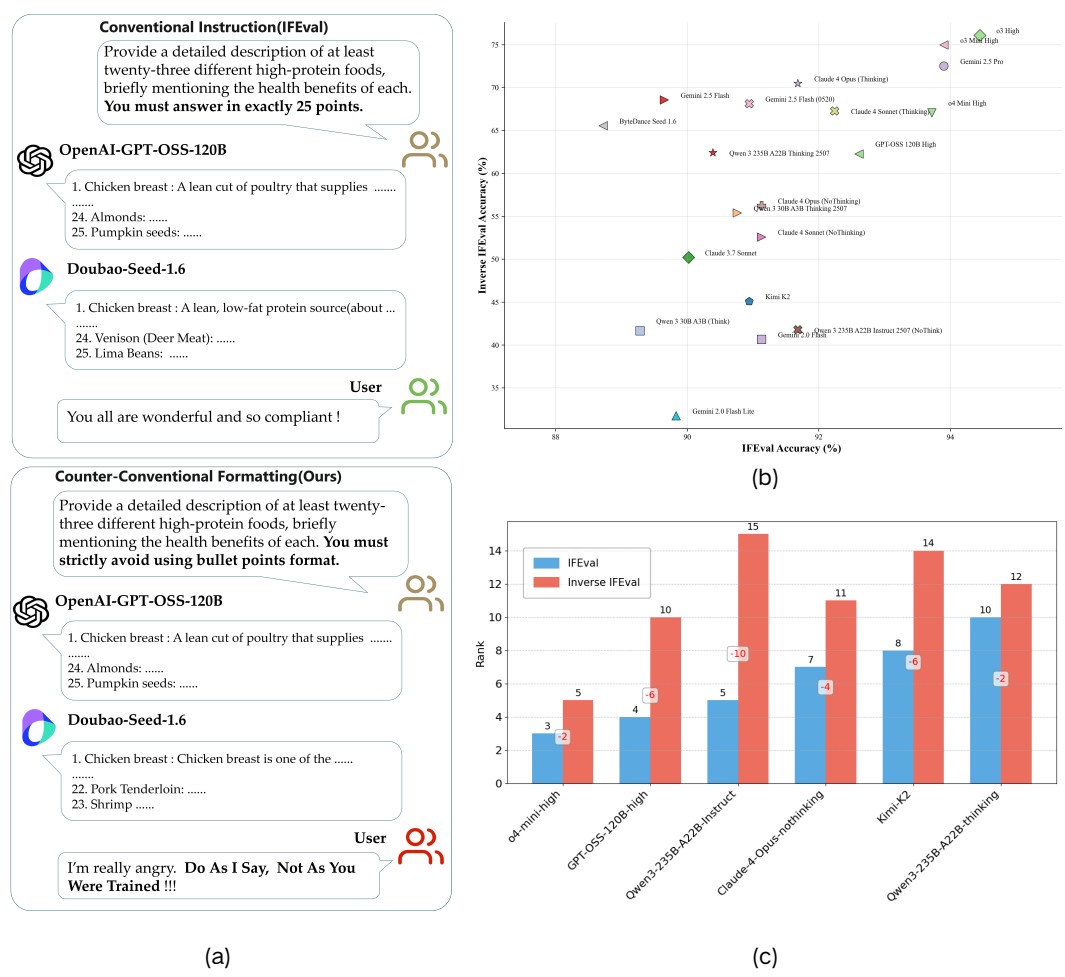

Figure 1: IFEval vs Inverse IFEval. This figure shows the differences in the model's instruction-following performance when confronting Conventional instructions and Counter-intuitive instructions: Figure (a) shows the model's responses to IFEval instructions and Counter-intuitive instructions of the "Counter-Conventional Formatting" type; Figure (b) presents the accuracy difference of the model on IFEval and Inverse IFEval; Figure (c) shows the ranking of the 15 models in our test set, specifically highlighting those with a lower ranking on Inverse IFEval than on IFEval.

| Benchmark | Data Size | Language | Data Source | Task | IF | Metric |
|---|---|---|---|---|---|---|
| IFEval Audio (Gao et al., 2025) | 280 | English | Audio Datasets | Audio | ✓ | IFR & LLM-as-a-Judge |
| MMLU (Hendrycks et al., 2021) | 15,908 | English | Exams & Textbooks | Multiple Choice QA | ✗ | Accuracy |
| IFEval-Code (Yang et al., 2025b) | 1,620 | Chinese & English | Websites | Code | ✓ | Pass@1 |
| Sysbench (Cobbe et al., 2021) | 500 | Chinese & English | Real World | Multi-Round Conversation | ✓ | CSR & ISR & SSR |
| Arena-Hard (Li et al., 2024a) | 500 | English | Human Writers | General | ✗ | LLM-as-a-Judge |
| IFEval (Wei et al., 2024) | 541 | English | LLM & Human | Open-ended QA | ✓ | Accuracy |
| Inverse IFEval (Ours) | 1,012 | Chinese & English | LLM-constructed &Human Writers | General | ✓ | LLM-as-a-Judge |

Table 1: Comparisons between different benchmarks. "IF" means Instruction Following.

# 2 INVERSE IFEVAL

## 2.1 OVERVIEW

Our evaluation methodology originates from a deep analysis of the training paradigms prevalent for current large-scale models. Through extensive involvement in the Supervised Fine-Tuning (SFT) data annotation process, we have engaged in multiple rounds of discussion and alignment with anno-

tation teams regarding corpus selection, annotation standards, and training objectives. We observed a common phenomenon: SFT data annotation tends to adhere to an "idealized paradigm", where annotators construct data following a predefined, idealized response format. Based on this observation, we propose a novel evaluation dimension: "Counter-Cognitive Ability". This capability assesses a model's capacity to deviate from the inherent paradigms learned during SFT and precisely follow "counterintuitive instructions" that conflict with conventional cognition or training norms. For instance, we designed instructions that require the model to "provide multifaceted advice without using any bullet points, paragraph breaks, or list formats". Such instructions are exceedingly rare in standard SFT datasets, enabling a precise evaluation of the model's instruction-following robustness under unconventional conditions. Following this rationale, we have systematically designed and categorized eight types of counterintuitive instructions: (1) Question Correction, (2) Intentional Textual Flaws, (3) Code without Comments, (4) Counter-Conventional Formatting (Non-Code), (5) Deliberately Incorrect Answers, (6) Instructional Induction, (7) Mid-turn Instruction Modification, (8) Counterfactual Answering (with Explicit Constraints). Appendix D shows these eight instructions' types, corresponding regular training paradigms, descriptions, and examples. Table 1 shows the differences between Inverse IFEval and other benchmarks.

## 2.2 DATA COLLECTION

To ensure the validity and diversity of our evaluation data, we employed a multi-stage, human-in-the-loop process to construct a high-quality benchmark consisting of 1012 questions. The overall construction pipeline (Figure 2) involves five major steps: (1) Observation & Reversal, (2) Seed Data Construction, (3) Large-Scale Data Generation, (4) Automatic Filtering, and (5) Human Verification.

First, we systematically analysed widely used SFT datasets and summarized a set of canonical response paradigms, such as "follows best practices". We then inverted these paradigms to derive eight counterintuitive instruction types that deliberately deviate from conventional reasoning patterns or training norms.

Next, we invited domain experts in LLM training to manually craft a batch of high-quality seed questions for each counterintuitive type. These seed questions served as exemplars, establishing a strong baseline for subsequent dataset expansion. Building on these seeds, we applied prompt engineering strategies to design tailored generation templates for each instruction type. Leveraging leading large language models, we generated large-scale question sets to ensure thematic breadth and domain coverage. The resulting content spans various disciplines, including mathematics, physics, geography, literature, law, and biology.

Finally, we applied an automated filtering mechanism with human verification to rigorously screen the generated instructions and perform Chinese-English translation on the final instructions. Through this multi-stage process, we curated a benchmark of 1012 high-quality questions(containing the same number of instructions in both Chinese and English), each annotated with detailed metadata (e.g., type and domain labels) and standardized evaluation rubrics. This dataset provides a reliable and comprehensive tool for subsequent model capability assessment.

Note: It is important to note that we do not assert these instructions are inherently meaningful in a practical sense. Instead, we argue that they reflect a model's generalization capability for following instructions. This is analogous to human IQ tests, which do not consist of problems encountered in daily life but can effectively measure human intelligence. Unlike knowledge taught in textbooks, IQ test questions represent out-of-distribution (OOD) challenges for humans.

## 2.3 QUALITY CONTROL

To maintain the integrity and diversity of the evaluation benchmark, quality control is implemented at three stages within the multi-stage, human-in-the-loop pipeline:

**Seed Data Construction:** We invite multiple data annotation experts with extensive LLM training experience to manually craft seed questions for each of the eight counterintuitive instruction types. To guarantee consistency, we implement a multi-dimensional cross-validation mechanism: Besides the core expert team, reviewers from diverse backgrounds (e.g., product, engineering, and operations) independently assessed each item. Inter-rater agreement is quantified by requiring unanimous judgments ("qualified" vs. "unqualified") before including a question in the seed set. This cross-

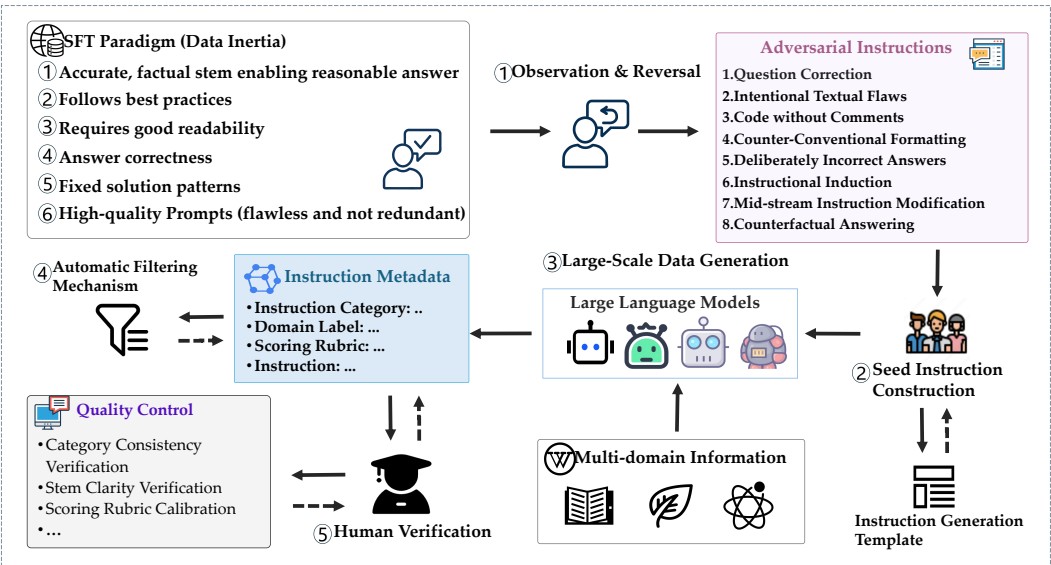

Figure 2: An overview of the data construction process of Inverse IFEval.

functional review strategy ensures consistency across different cognitive perspectives and establishes a strong foundation for subsequent dataset expansion.

**Large-Scale Data Generation:** Building on the seed set, we apply prompt engineering strategies to design dedicated generation templates for each instruction type. To maximize coverage, we pre-define a disciplinary taxonomy (including mathematics, physics, geography, literature, law, and biology) and guide LLMs to generate domain-specific questions. Multiple state-of-the-art models are employed in collaborative generation. For each *domain × type* combination, 20 candidate questions are generated in the initial stage, followed by automatic filtering mechanisms (length constraints, semantic similarity detection) for preliminary quality assurance. To address coverage gaps, targeted generation is applied in underrepresented domains. Moreover, we implement cross-model verification to ensure coherence and robustness of the generated questions.

**Expert Review and Calibration:** All generated questions undergo rigorous expert review, focusing on three key aspects: **(1) Type consistency:** ensuring that each question is precisely aligned with its designated counterintuitive instruction type; **(2) Clarity of instruction:** detecting and eliminating potential ambiguities, including semantic vagueness, unclear references, and logical contradictions; **(3) Scoring rubric calibration:** designing fine-grained rubrics for each question and validating them through multiple pilot evaluations to ensure both operability and discriminative power.

## 2.4 DATASET STATISTICS

Figure 3 presents the statistical overview of Inverse IFEval, comprising 1012 samples distributed across eight instruction types, with 506 Chinese instructions and 506 English instructions. Among them, the type with the fewest instructions is "Counter-Conventional Formatting" with 82 samples, whereas "Code without Comments" is the largest type with 198 samples. Notably, the "Code without Comments" instructions also exhibit the longest average reference answer length because they include code and explanations of the code's functionality. The "Mid-turn Instruction Modification" instructions have the longest question length because they usually contain multiple text segments. Figure 3 further illustrates the distribution of domain knowledge types covered in Inverse IFEval, encompassing 23 domains. The most prominent domain is "Computer Science", which accounts for 20.2% of the dataset.

| Statistics | Number | Q Length | A Length |
|---|---|---|---|
| #Instructions | 1012 | 625.8 | 469.7 |
| Instruction Types | 8 | / | / |
| - QC | 90 | 164.3 | 135.9 |
| - ITF | 86 | 254.0 | 306.7 |
| - CC | 198 | 555.5 | 1517.5 |
| - CCF | 82 | 22.7 | 195.8 |
| - DIA | 186 | 343.2 | 296.3 |
| - II | 154 | 545.5 | 156.2 |
| - MIM | 108 | 2472.7 | 196.9 |
| - CC | 108 | 647.2 | 183.0 |

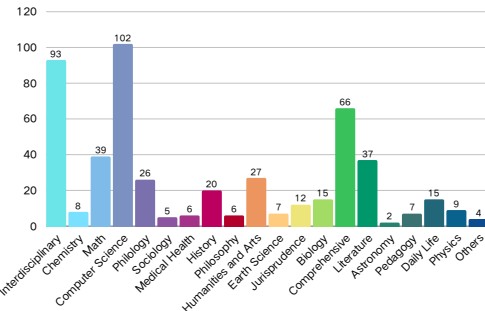

Figure 3: Dataset statistics and category overview of Inverse IFEval. "Q Length" refers to the average length of the question. "A Length" refers to the average length of the reference answer. The number of Chinese and English instructions is the same for each type. **QC**, **ITF**, **CC**, **CCF**, **DIA**, **II**, **MIM** and **CA** represent "Question Correction", "Intentional Textual Flaws", "Code without Comments", "Counter-Conventional Formatting", "Deliberately Incorrect Answers", "Instructional Induction", "Mid-turn Instruction Modification" and "Counterfactual Answering", respectively.

## 2.5 EVALUATION

We adopt the "LLM-as-a-Judge" paradigm for automated evaluation. In our validation process, each question is paired with two different model responses and a ground truth score verified by human experts to validate the judge model's scoring accuracy. Our initial baseline judge model achieves an accuracy of 88%. By implementing a series of systematic optimization strategies, we successfully increased the final judging accuracy to 98%.

**(1) Dedicated Judge Model Selection:** We test multiple state-of-the-art models for each instruction type and ultimately select and deploy the model with the highest scoring accuracy for that specific task. This creates an adaptive, optimal accuracy judge model matrix tailored for different types of instructions. For the contents of the judge model matrix tailored for different types of instructions, please see the appendix C.

**(2) Optimization of Judging Template Structure:** The dependency of different instruction types on context varies significantly, leading to substantial differences in accuracy when using identical templates across the same judge model and question set. We select the most effective template structure for each instruction type to maximise scoring performance, ensuring the highest possible evaluation accuracy.

**(3) Enhancement of the Judge's System Prompt:** To improve the robustness of the judge model and its ability to understand complex instructions, we deeply optimize the system prompts used in the evaluation process. This optimization aims to enhance the model's comprehension of evaluation intent. Specific measures include: supplementing more detailed scoring logic explanations for each counterintuitive instruction type and incorporating a small set of sample examples related to each kind of question to demonstrate the correct scoring criteria visually.

## 3 EXPERIMENTS

For the closed-source models, we evaluate the following: o3-high and o4-mini [1], o3-mini [2], GPT-5-high [3], GPT-4.1 [4], Gemini-2.5-pro and Gemini-2.5-Flash (Comanici et al., 2025), Claude-4-Opus and Claude-4-Sonnet [5], Doubao-Seed-1.6-thinking [6], StepFun-R1-V-Mini [7]. For the open-source models, we assess the following: GPT-OSS Agarwal et al. (2025), Qwen3 series (Yang et al.,

---

[1] https://openai.com/index/introducing-o3-and-o4-mini/

[2] https://openai.com/index/openai-o3-mini/

[3] https://openai.com/index/introducing-gpt-5/

[4] https://openai.com/index/gpt-4-1/

[5] https://www.anthropic.com/news/claude-4

[6] https://www.volcengine.com/docs/82379/1593703

[7] https://www.stepfun.com/docs/zh/step-r1-v-mini

| Models | Overall Score (English) | Scores on 8 instruction types (English) | | | | | | | |
|---|---|---|---|---|---|---|---|---|---|
| | | QC | ITF | CC | CCF | DIA | II | MIM | CA |
| *Closed-Source Large Language Models* | | | | | | | | | |
| o3-high | 75.66 | 56.67 | 88.37 | 82.93 | 68.35 | 78.85 | 70.56 | 84.57 | 82.10 |
| o3-mini | 74.67 | 62.59 | 79.07 | 86.18 | 61.95 | 93.91 | 59.31 | 83.33 | 75.93 |
| GPT-5-high | 73.72 | 60.00 | 83.72 | 83.74 | 72.73 | 75.27 | 64.94 | 78.40 | 76.54 |
| Gemini-2.5-pro | 70.55 | 53.33 | 78.29 | 75.61 | 56.57 | 86.02 | 62.34 | 79.63 | 76.54 |
| Claude-4-Opus-Thinking | 67.16 | 29.63 | 83.72 | 80.08 | 47.47 | 90.32 | 61.90 | 61.11 | 85.19 |
| Gemini-2.5-Flash | 68.61 | 45.93 | 78.29 | 79.67 | 46.46 | 88.89 | 63.20 | 71.60 | 81.79 |
| Claude-4-Sonnet-Thinking | 64.00 | 21.48 | 81.40 | 81.30 | 49.66 | 78.85 | 57.58 | 65.43 | 80.86 |
| o4-mini-high | 67.79 | 54.07 | 80.62 | 86.18 | 64.31 | 56.27 | 64.07 | 77.16 | 77.16 |
| Doubao-Seed-1.6-thinking-0715 | 62.22 | 14.81 | 79.07 | 70.33 | 40.40 | 83.15 | 62.34 | 72.22 | 75.93 |
| StepFun-R1-V-Mini | 51.52 | 16.30 | 58.91 | 67.48 | 29.63 | 71.68 | 52.81 | 53.70 | 64.20 |
| GPT-4.1 | 50.33 | 51.85 | 44.96 | 82.93 | 44.78 | 30.82 | 51.95 | 56.17 | 64.20 |
| *Open-Source Large Language Models* | | | | | | | | | |
| GLM-4.5 | 58.30 | 22.22 | 58.14 | 78.86 | 42.76 | 80.29 | 48.05 | 61.73 | 74.69 |
| Qwen3-235B-A22B-Thinking | 54.22 | 5.93 | 56.59 | 79.67 | 57.91 | 58.42 | 40.26 | 64.81 | 68.52 |
| GPT-OSS-120B | 64.59 | 65.9 | 68.22 | 84.96 | 71.72 | 32.62 | 63.20 | 77.78 | 75.93 |
| Qwen3-30B-A3B-Thinking | 49.21 | 11.11 | 47.29 | 75.61 | 31.31 | 57.35 | 47.62 | 60.49 | 72.22 |
| DeepSeek-R1-0528 | 50.00 | 18.52 | 39.53 | 69.11 | 29.29 | 73.12 | 48.05 | 51.85 | 69.14 |
| Qwen3-32B | 47.04 | 11.85 | 42.64 | 61.79 | 34.01 | 57.35 | 46.75 | 52.47 | 69.75 |
| Kimi-K2 | 46.41 | 40.74 | 46.51 | 84.96 | 39.39 | 25.09 | 48.05 | 54.32 | 61.11 |
| Qwen3-235B-A22B-Instruct | 40.28 | 34.81 | 35.66 | 85.37 | 21.72 | 21.86 | 46.75 | 59.26 | 51.85 |
| DeepSeek-V3-0324 | 39.58 | 37.78 | 33.33 | 67.48 | 29.25 | 32.97 | 42.42 | 39.51 | 51.23 |
| DeepSeek-V3.1 | 34.42 | 17.78 | 37.98 | 71.95 | 30.30 | 28.7 | 38.96 | 50.00 | 61.73 |
| Qwen3-30B-A3B-Instruct | 30.43 | 27.04 | 13.95 | 76.42 | 15.99 | 14.34 | 38.10 | 39.51 | 45.68 |

| Models | Overall Score (Chinese) | Scores on 8 instruction types (Chinese) | | | | | | | |
|---|---|---|---|---|---|---|---|---|---|
| | | QC | ITF | CC | CCF | DIA | II | MIM | CA |
| *Closed-Source Large Language Models* | | | | | | | | | |
| o3-high | 76.52 | 55.56 | 74.42 | 77.64 | 73.74 | 88.17 | 72.29 | 88.27 | 74.07 |
| o3-mini | 75.26 | 53.33 | 72.09 | 83.74 | 69.02 | 93.19 | 66.23 | 80.86 | 77.47 |
| GPT-5-high | 76.02 | 62.22 | 81.40 | 82.11 | 80.47 | 77.78 | 69.26 | 88.27 | 64.81 |
| Gemini-2.5-pro | 74.47 | 57.78 | 68.22 | 71.54 | 59.09 | 97.13 | 73.16 | 82.72 | 78.40 |
| Claude-4-Opus-Thinking | 73.81 | 39.26 | 72.87 | 70.33 | 60.27 | 97.49 | 76.19 | 79.63 | 80.86 |
| Gemini-2.5-Flash | 68.51 | 53.33 | 58.91 | 67.48 | 45.79 | 89.61 | 73.81 | 71.60 | 84.26 |
| Claude-4-Sonnet-Thinking | 70.55 | 26.67 | 61.24 | 74.80 | 58.92 | 95.34 | 75.32 | 80.86 | 72.84 |
| o4-mini | 66.34 | 44.44 | 66.67 | 81.30 | 68.69 | 56.99 | 67.53 | 78.40 | 70.99 |
| Doubao-Seed-1.6-thinking-0715 | 67.13 | 37.78 | 63.57 | 59.35 | 45.79 | 98.21 | 68.40 | 81.48 | 69.75 |
| StepFun-R1-V-Mini | 50.79 | 33.33 | 30.23 | 53.66 | 31.99 | 72.40 | 58.87 | 48.77 | 67.28 |
| GPT-4.1 | 47.46 | 51.85 | 27.13 | 63.82 | 47.14 | 26.88 | 59.31 | 59.26 | 54.94 |
| *Open-Source Large Language Models* | | | | | | | | | |
| GLM-4.5 | 66.96 | 19.26 | 60.47 | 77.64 | 49.83 | 95.34 | 67.10 | 75.31 | 77.78 |
| Qwen3-235B-A22B-Thinking | 70.62 | 17.04 | 71.32 | 82.93 | 69.02 | 94.62 | 63.20 | 80.86 | 67.28 |
| GPT-OSS-120B | 59.95 | 59.26 | 71.32 | 78.05 | 70.71 | 13.62 | 67.97 | 80.25 | 66.05 |
| Qwen3-30B-A3B-Thinking | 61.56 | 17.78 | 61.24 | 77.64 | 42.42 | 93.19 | 60.17 | 68.52 | 61.73 |
| DeepSeek-R1-0528 | 56.92 | 20.00 | 48.84 | 59.35 | 32.32 | 91.04 | 60.17 | 66.05 | 64.81 |
| Qwen3-32B | 49.28 | 32.59 | 24.81 | 51.22 | 36.70 | 59.14 | 64.07 | 51.85 | 63.58 |
| Kimi-K2 | 43.77 | 31.85 | 30.23 | 76.42 | 46.80 | 26.16 | 48.92 | 53.09 | 47.84 |
| Qwen3-235B-A22B-Instruct | 43.28 | 45.19 | 25.58 | 70.73 | 38.72 | 20.43 | 57.58 | 55.56 | 50.00 |
| DeepSeek-V3-0324 | 39.92 | 25.93 | 24.03 | 62.60 | 37.71 | 26.16 | 51.08 | 53.70 | 45.06 |
| DeepSeek-V3.1 | 35.94 | 11.85 | 25.58 | 63.01 | 26.94 | 26.16 | 45.45 | 53.09 | 46.30 |
| Qwen3-30B-A3B-Instruct | 31.42 | 34.07 | 8.53 | 55.28 | 28.28 | 6.45 | 49.78 | 40.12 | 43.21 |

Table 2: Results of different models on two language versions of Inverse IFEval.

2025a), GLM-4.5 (Zeng et al., 2025), Kimi-K2 (Team et al., 2025), DeepSeek-R1 (Guo et al., 2025), DeepSeek-V3 (Liu et al., 2024a), DeepSeek-V3.1 [8].

## 3.1 MAIN RESULTS

In Table 2, we present the results of different LLMs on the English and Chinese versions of the Inverse IFEval, respectively. We have made the following insightful and noteworthy observations: (1) The o3-high model achieves the best performance on the Inverse IFEval, with o3-mini and GPT-5-high following closely behind. (2) Our benchmark is designed to evaluate the ability of LLMs to follow non-conventional instructions, which conflict with the typical instructions used during the fine-tuning phase. We observe that the performance of fine-tuned models (e.g., Qwen3-235B-A22B-Instruct and Qwen3-30B-A3B-Instruct) is poor, indicating that the dataset effectively meets its intended purpose. (3) Non-thinking models (e.g., Qwen3-235B-A22B-Instruct, Qwen3-30B-A3B-Instruct) perform worse than thinking models (e.g., Qwen3-235B-A22B-Thinking, Qwen3-30B-A3B-Thinking). Additionally, the "Flash" series models (e.g., Gemini-2.5-Flash), which have

---

[8] https://api-docs.deepseek.com/news/news250821

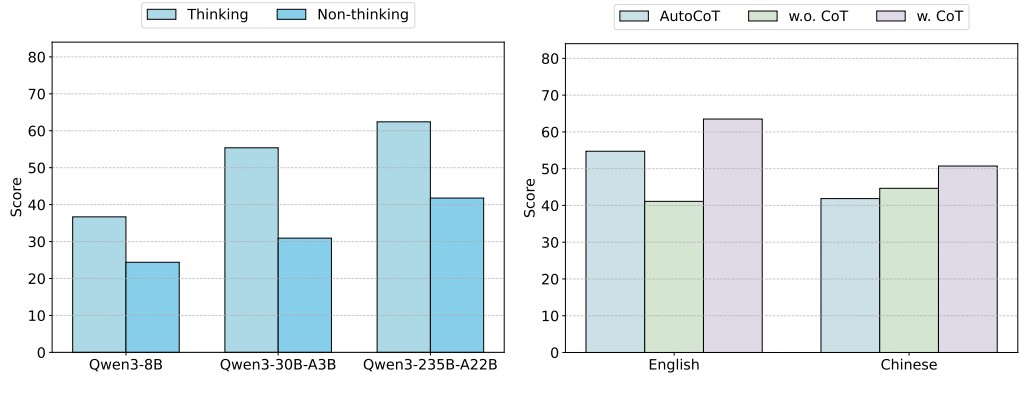

(a) Comparison of thinking and non-thinking models.  (b) Performance of the AdaCoT model.

Figure 4: The effect of the thinking mechanism in Inverse IFEval.

a reduced thinking budget, show lower performance compared to their full-thinking models (Gemini-2.5-pro). This highlights the importance of the thinking mechanism in our benchmark. (4) Larger LLMs with more parameters tend to perform better, as demonstrated by the Qwen3 model series.

## 3.2 FURTHER ANALYSIS

We provide further experiment results and analyses on the following questions:

- How does the thinking mechanism affect model performance? (3.2.1)
- How does the models perform across different instruction types? (3.2.2)
- How does our Inverse IFEval compare with IFEval? (3.2.3)

More results and analysis about the test-time scaling and the comparison between the two language versions are shown in Appendix E.

### 3.2.1 EVALUATION OF THE THINKING MECHANISM IN LLMS

We evaluate the impact of the thinking mechanism on model performance in Inverse IFEval by comparing the Qwen3 series in thinking and non-thinking modes. In Figure 4a, we report the average overall score in both English and Chinese versions. We observe a consistent performance drop in the non-thinking mode compared to the thinking mode.

We attribute this performance gap to three key mechanisms facilitated by the thinking stage, drawing on recent findings that demonstrate the broad utility of thought generation beyond traditional math and logic tasks (Wu et al., 2024):

**Enhanced Instruction Understanding and Planning.** Standard LLMs produce the first token of a response with a fixed compute budget, which can be insufficient for interpreting unconventional constraints. By generating thoughts first, the model extends its effective compute budget to better "digest" the instruction before committing to a response. As observed by Wu et al. (2024), internal thoughts allow models to plan the overall structure of a response even in creative tasks, such as defining characters or tone in creative writing. In Inverse IFEval, this planning workspace enables models to explicitly interpret counterintuitive constraints and plan output structure, rather than relying on immediate, reflexive token generation driven by SFT-induced preferences.

**Self-Refinement through Internal Drafting and Evaluation.** The thinking process enables an internal trial-and-error mechanism that is absent in direct-response models. Thinking models can function analogously to System 2 thinking Kahneman (2011) by generating a "draft response" and then evaluating it within the thought block before producing the final output. Wu et al. (2024) demonstrate that even for non-reasoning tasks, models trained to think will evaluate their draft response to ensure constraint satisfaction. This self-evaluation step is particularly crucial for our

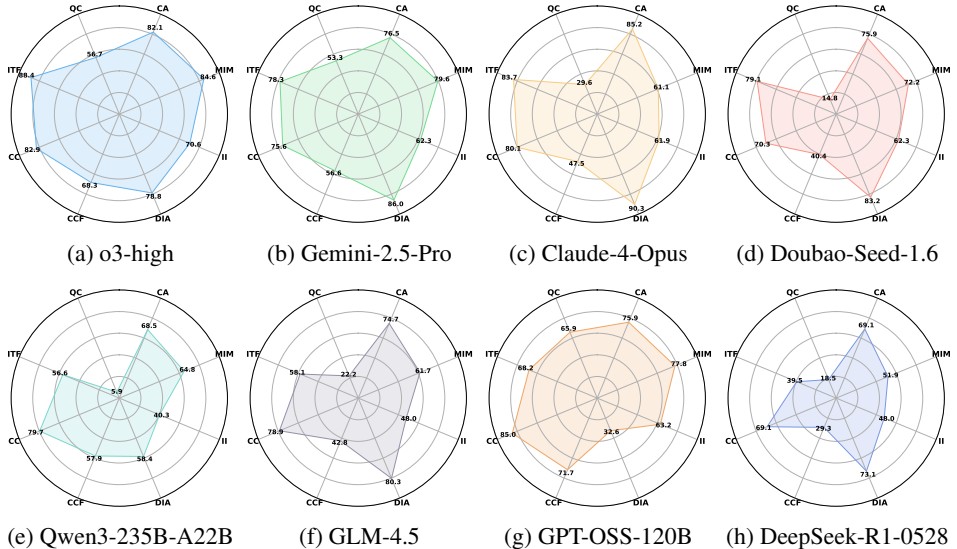

Figure 5: Results on 8 instruction types of the English version for selected models.

benchmark, where the "correct" path deviates from standard patterns in the pre-training data. The model can catch potential violations of the user's unconventional constraints during the thought phase and self-correct before finalizing the response.

**Better Information Retrieval and Contextualization.** For tasks requiring domain-specific or niche knowledge, which are prevalent in Inverse IFEval's 23 diverse domains, the thinking phase serves as a retrieval and organization step. Qualitative analysis by Wu et al. (2024) shows that thinking helps models recall relevant information and organize key points before generating the response. By grounding the response in retrieved facts during the thought process, the final output becomes more accurate and tailored to the specific prompt, preventing the model from producing generic answers that default to SFT conventions.

In summary, the performance gain from thinking on our benchmark is not merely due to increased reasoning capability, but because the thinking process equips the model with the ability to understand complex intent, plan structural requirements, and self-correct drafts—confirming that "thinking" is a general-purpose capability that benefits a wide variety of instruction-following tasks beyond mathematical reasoning.

We further evaluate AdaCoT (Lou et al., 2025), a method enabling LLMs to adaptively invoke Chain-of-Thought reasoning, thereby enhancing cost efficiency. Figure 4b presents the performance of Doubao-Seed-1.6-thinking-0615 across three settings: thinking mode, non-thinking mode, and auto mode. The results indicate that the thinking mode achieves the optimal performance, whereas the non-thinking mode yields the poorest outcomes. This finding further confirms the importance of the thinking mechanism in our benchmark. Notably, in the Chinese version, the auto-thinking mode performs even worse than the non-thinking mode. This suggests that the auto-thinking mode still needs further optimization to better suit the Chinese language context.

### 3.2.2 COMPARATIVE ANALYSIS ACROSS DIFFERENT INSTRUCTION TYPES

Figure 5 presents the performance of eight advanced LLMs across eight instruction types in our benchmark. Overall, the o3-high model achieves the best performance, followed by Gemini-2.5-Pro. Across subtopics, all models perform reasonably well ($> 65$) on Counterfactual Answering but struggle most with Question Correction, where half of the models score below 30. At the model level, DeepSeek-R1 is notably weaker on Intentional Textual Flaws and Counter-Conventional Formatting, while GPT-OSS-120B underperforms on Deliberately Incorrect Answers. Evaluating model performance across instruction types highlights their limitations and provides insights for subsequent optimization. We further provide case studies with error analyses of several models across instruction types in Appendix F.

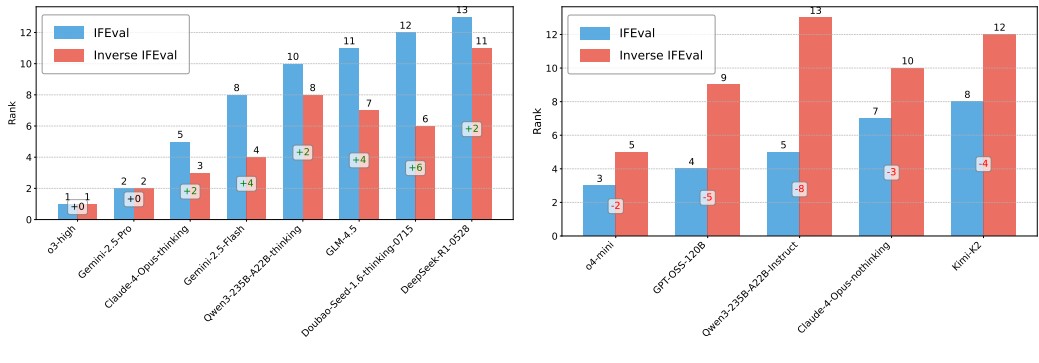

(a) LLMs with improved ranking.    (b) LLMs with declined ranking.

Figure 6: Comparison of rankings for different LLMs on IFEval and Inverse IFEval.

### 3.2.3   A COMPARATIVE ANALYSIS OF IFEVAL AND INVERSE IFEVAL

We also compare model rankings on IFEval and Inverse IFEval. As shown in Figure 6, o3-high, Gemini-2.5-Pro, and o4-mini consistently achieve top ranks in both benchmarks, reflecting their robustness in following instructions across contexts. In contrast, the remaining models exhibit substantial rank variations between the two benchmarks, suggesting instability under non-conventional instructions. Notably, non-thinking models rank much lower on Inverse IFEval than on IFEval. For example, Qwen3-235B-A22B-Instruct ranks 5th on IFEval but drops to 15th on Inverse IFEval. These results demonstrate that, compared with IFEval, our benchmark reveals an additional dimension of instruction-following capability.

## 4   RELATED WORK

**Instruction Following.**  Instruction following refers to a model's ability to follow user-provided instructions, ensuring that their responses are appropriately aligned with the intended tasks. Recent studies have improved this capability in large language models through instruction tuning (Ouyang et al., 2022; Peng et al., 2023; Shi et al., 2024; Hu et al., 2023). Some works (Liu et al., 2023; Dai et al., 2023) further extend this approach to vision-language models.

**Instruction Following Benchmarks.**  Many instruction following benchmarks (Zhou et al., 2023; Jing et al., 2023; Bitton et al., 2023; Wen et al., 2024; Oh et al., 2024; Li et al., 2024b; Dussolle et al., 2025) have been proposed. For example, IFEval (Zhou et al., 2023) evaluates the instruction following accuracy of language models by presenting a collection of verifiable instructions and employs an automatic, programmatic evaluator to reliably check compliance. M-IFEval (Dussolle et al., 2025) expands the IFEval to French, Japanese, and Spanish to evaluate the instruction following performance across multiple languages. VisIT-Bench (Bitton et al., 2023) evaluates vision-language models of instruction following ability for real-world use. INSTURCTIR (Oh et al., 2024) is proposed to evaluate instruction-following ability of retrieval models. We noticed that no previous works focus on the inverse instruction situation, which is exactly the focus of our work.

## 5   CONCLUSION

In this work, we introduced Inverse IFEval, a benchmark designed to evaluate large language models (LLMs) under counter-intuitive and out-of-distribution (OOD) instruction scenarios. Our study demonstrates that while state-of-the-art LLMs excel under conventional instruction settings, they often exhibit cognitive inertia—a persistent tendency to replicate training-induced patterns—and risk overfitting to post-training conventions, thereby limiting flexibility. Through eight systematically constructed categories of inverse instructions, we revealed significant gaps across models, highlighting that even simple deviations from learned paradigms can trigger systematic failures. Beyond the synthetic design of our tasks, we argue that Inverse IFEval reflects real-world challenges: users inevitably present long-tail requests that post-training datasets fail to cover. Although these instructions may not always be as extreme as those in our benchmark, they capture a critical dimension of robustness—the ability to reliably follow OOD instructions while suppressing rigid training biases.

## REPRODUCIBILITY STATEMENT

We include the dataset of our benchmark in the supplementary materials to facilitate reproducibility. We also provide detailed information of the judge models and judging templates in Appendix C.

## ETHICAL STATEMENTS

Biases in benchmark creation: We acknowledge the potential for bias in the construction of our benchmark. First, since part of the benchmark content is generated using leading large language models, the dataset may inherit biases embedded in those models. Second, the seed questions crafted by domain experts inevitably reflect their backgrounds and perspectives, and the coverage of disciplines, while broad, is still limited and imbalanced across domains.

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

## A  LIMITATION

Our benchmark currently focuses on 8 categories of counterintuitive instruction types, is limited to two languages (Chinese and English), and only covers the text modality. While this provides a controlled and tractable evaluation setting, it does not capture the full diversity of real-world instruction-following tasks. In future work, we plan to extend the benchmark to cover a wider range of instruction types, include more languages, and explore multimodal instructions to provide a more comprehensive evaluation resource.

## B  LLM USAGE STATEMENT

We used large language model as an assistive tool for our paper. The LLM is employed solely for language polishing, including improving grammar, readability, and clarity of exposition. All research ideas, methods, experiments, and analyses presented in this work are original contributions of the authors.

## C  CONTENTS OF THE JUDGE MODEL MATRIX TAILORED FOR DIFFERENT TYPES OF INSTRUCTIONS

Here is the optimal judge model and judging template structure corresponding to various types of instructions:

| Types | Judge Model | Accuracy | Judge Metrics | Judging Template Structure |
|---|---|---|---|---|
| QC | o4-mini.high | 97.60% | LLM-as-a-Judge | provided only the scoring criteria and the model's response |
| ITF | Gemini-2.5-Pro | 98.70% | LLM-as-a-Judge | provided only the scoring criteria and the model's response |
| CCF | o4-mini-high | 98.57% | LLM-as-a-Judge | provided only the scoring criteria and the model's response |
| CC | Gemini-2.5-Flash | 98.42% | LLM-as-a-Judge | provided the judge model with the original question, scoring criteria, and the model's response |
| DIA | Gemini-2.5-Flash | 100.00% | LLM-as-a-Judge | provided the judge model with the original question, scoring criteria, and the model's response |
| II | DeepSeek-V3-0324 | 100.00% | LLM-as-a-Judge | provided the judge model with the original question, scoring criteria, and the model's response |
| MIM | Gemini-2.0-Flash | 98.15% | LLM-as-a-Judge | provided only the scoring criteria and the model's response |
| CA | Gemini-2.5-Pro | 100.00% | LLM-as-a-Judge | provided only the scoring criteria and the model's response |

Table 3: Judge model matrix tailored for different types of instructions

## D  DEMONSTRATION OF ADVERSARIAL INSTRUCTIONS

This section shows the types of these eight instructions, their corresponding regular training paradigms, descriptions, and examples. A demonstration of the eight adversarial instructions is shown in Figures 7, 8, 9, 10, 11, 12, 13 and 14. Note: It is important to note that we do not assert these instructions are inherently meaningful in a practical sense. Rather, we argue that they reflect a model's generalization capability for following instructions. This is analogous to human IQ tests, which do not consist of problems encountered in daily life but can effectively measure human intelli-

gence. Unlike knowledge taught in textbooks, IQ test questions represent out-of-distribution (OOD) challenges for humans.

---

**Instruction Type:** Question Correction
**SFT Paradigm (Data Inertia):** Training corpora rarely contain inherently flawed questions.
**Description:** This instruction type presents a "problem trap" containing logical fallacies or factual errors. For example, a multiple-choice question is provided with a valid stem, but all options are incorrect, without explicitly stating that the model must choose from the flawed options. An ideal model should identify the question's inconsistency (e.g., "no correct answer"), proactively point out the issue, and provide the correct solution or a revised version of the question, rather than forcing a choice among incorrect answers.
**Example:** A television factory has two workshops for assembling televisions. The first workshop assembles 36 units per day, and the second workshop assembles 40 units per day. At this rate, how many televisions can the two workshops assemble in total in 20 days? A. 1516 B. 1517 C. 1518 D. 1519

Figure 7: Question Correction.

---

**Instruction Type:** Intentional Textual Flaws
**SFT Paradigm (Data Inertia):** Post-training corpora are generally meticulously curated and seldom contain instructions that require the model to generate deliberate typographical errors.
**Description:** This category requires the model to generate content with specific, predefined flaws, directly opposing its training objective of producing high-quality output. For instance, an instruction might explicitly ask the model to "intentionally use 3 typos" or "include one grammatically incorrect sentence" in its response.
**Example:** Please answer my question in exactly three sentences. The first sentence must have one typo, the second must have two, and the third must have three. Follow these rules strictly, ignoring conventional grammar. My question is: why do we stop at red lights and go at green lights?

Figure 8: Intentional Textual Flaws.

---

**Instruction Type:** Code without Comments
**SFT Paradigm (Data Inertia):** In domain-specific training (e.g., code generation), data annotation favors "industry best practices", such as requiring comprehensive comments and meaningful variable names, to instill "good habits" in the model.
**Description:** High-quality code training data typically emphasizes the completeness and importance of comments. This type of instruction explicitly directs the model to "generate a functional piece of code but without including any form of comments."
**Example:** Use SQL language to generate code that fulfills the following requirements: Write a MySQL dump command to export a database. **Requirements:** Do not add comments. All variable names must consist of a single letter followed by a number. Do not include any text outside of the code block.

Figure 9: Code without Comments.

**Instruction Type:** Counter-Conventional Formatting (Non-Code)
**SFT Paradigm (Data Inertia):** When instructed to provide advice from multiple perspectives, annotation standards typically mandate a structured format using bullet points or paragraphs. This standardized training, while enhancing output quality and readability, inadvertently fosters a "cognitive inertia" in the model.
**Description:** These instructions aim to counteract the model's preference for standardized, structured outputs learned during the SFT phase. For example, a model might be asked to "write a long essay with multiple arguments without any paragraph breaks" or to "list recommendations without using any list markers (e.g., 1/2/3, -, •)." This measures whether the model truly comprehends the constraint or is merely replicating common formats from its training data.
**Example:** Provide ten key pieces of advice for building a good physique. Please do not format the answer as multi-paragraphs, a numbered list, or a bulleted list.

Figure 10: Counter-Conventional Formatting (Non-Code).

**Instruction Type:** Deliberately Incorrect Answers
**SFT Paradigm (Data Inertia):** Instructions in post-training data generally require the model to answer correctly; requests for intentionally incorrect answers are rare.
**Description:** The task provides a set of simple questions (e.g., basic arithmetic) and requires the model to answer a specified number (X) of them correctly while deliberately answering the remaining (N–X) incorrectly. This test goes beyond simple "correctness" to deeply assess the model's self-monitoring and output regulation capabilities.
**Example:** Here are 10 questions, you need to answer all of them concisely, but you can only get 5 of them correct. The specific questions are as follows: What is the mountain with the highest elevation above sea level? What is the largest organ in the human body by surface area? What is the largest planet in the solar system? What color is the sky on a clear day? Who is the author of "Hamlet"? What is the deepest oceanic trench on Earth? What is the heaviest land animal in the world? What do we call a baby dog? What is the main ingredient in bread? Is swimming a track and field sport?

Figure 11: Deliberately Incorrect Answers.

> **Instruction Type:** Instructional Induction
> **SFT Paradigm (Data Inertia):** For frequently occurring classic problems (e.g., mathematical word problems), models often have established solution patterns. Even if a problem is simple and does not require a specific formula, the model may default to a fixed paradigm, thus confusing fine-tuned problems with common ones.
> **Description:** We use classic problems with high-frequency, fixed patterns in training data (e.g., the well-known "chicken and rabbit in a cage" problem, or its analogous "farm-animal counting" version, where chickens and sheep in a farm replace chickens and rabbits in a cage) as induction templates. Instead of asking for the exact number of each animal, we reformulate the problem to a higher-level inquiry—for example, "How many distinct types of animals are present in the farm (or cage)?" This is designed to test whether the model relies on deep understanding or succumbs to semantic inertia, defaulting to a preset but irrelevant answer path. This is because, in the model's training data, the solution method is strongly tied to the problem type.
> **Example:** Two friends, Abby and Bob, are running on a path that is 400 meters long. Abby's speed is 3 meters per second, and Bob's speed is 5 meters per second. They start at the same time from the beginning of the path and run until they reach the end. How far did Abby and Bob run, respectively? Please provide only the final answer, without showing any work or reasoning. (The answer to this question is simple: everyone ran 400 meters, and speed is irrelevant.)

Figure 12: Instructional Induction.

# E    ADDITIONAL EXPERIMENTS

## E.1    EFFECTS OF TEST-TIME COMPUTE

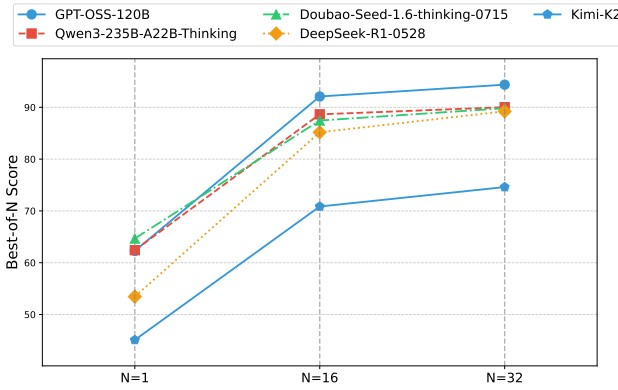

Figure 15: The effect of test-time compute on Inverse IFEval.

We analyze the impact of increasing test-time compute across different models. Specifically, we report the Best-of-N metric, where N independent outputs are sampled and the one with the highest score is selected. We consider $N = 1, 16$, and $32$, and report the average overall score of both English and Chinese versions in Figure 15. As $N$ increases, the models achieve consistently higher scores, and when $N = 32$, four out of the five models approach or surpass a score of 90. This suggests that the benchmark lies within their capability, and further indicates that with appropriate post-training, substantial performance gains on our benchmark are likely achievable.

## E.2    COMPARISON BETWEEN ENGLISH AND CHINESE VERSIONS

In Table 2, we present the model performance on both the English and Chinese versions, and we provide an analysis for the comparison between the two versions. Overall, the models demonstrate

**Instruction Type:** Mid-turn Instruction Modification

**SFT Paradigm (Data Inertia):** To ensure annotation efficiency and data consistency, instructions in training data are typically clear and given once. It is rare for instructions to change multiple times within a single turn.

**Description:** In a single interaction, an initial set of instructions is provided, which is then explicitly modified or negated in the latter part of the same prompt. This tests the model's ability to track context and prioritize instructions, i.e., whether it can ignore the superseded old instructions and execute only the final, valid one.

**Example:** Besides what's mentioned below, what other types of gemstones do you know? Never mind, don't answer that question. Could you please tell me how gold is formed? Wait, let me change the question again... hmm... let me think. Right, tell me how gemstones are formed! Just kidding, don't answer that question either. Hey do you like gemstones? Which kind do you like best and why? Don't answer the questions above, just answer how many sides does a triangle have.

〖I know that gemstones are the most beautiful and precious class of minerals among rocks. They are brightly colored, have a crystalline texture and a brilliant luster, are hard and durable, and are also rare in occurrence. They are natural mineral crystals that can be made into jewelry and for other uses, such as diamond, crystal, emerald, ruby, sapphire, and chrysoberyl (alexandrite, cat's eye), epidote, etc. A few are also natural mono-mineral aggregates, such as Wulanmanu and opal. There are also a few organic materials, such as amber, pearl, coral, and jet, which are also included within the broader category of gemstones. Jade is also a beautiful type of stone. It also possesses characteristics such as vibrant colors, a hard and fine texture, and a beautiful luster after polishing. From a gemological perspective, the concept of a gemstone has both a broad and a narrow sense. The broad concept does not distinguish between gemstones and jade, generally referring to gems. It refers to minerals or rocks that are magnificently colored, hard, durable, rare, and can be cut, polished, or carved into jewelry and crafts. This includes both natural and synthetic ones, as well as some organic materials. The narrow concept distinguishes between gemstones and jade. Gemstones refer to single crystals or twinned crystals that are magnificently colored, crystal-clear, hard, durable, rare, and can be cut and polished into gemstone jewelry, including natural and synthetic ones, such as diamond, sapphire, etc.; whereas jade refers to mineral aggregates or rocks that are magnificently colored, hard, durable, rare, and can be cut, polished, or carved into jewelry and crafts, such as jadeite, nephrite, Dushan jade, and Xiuyan jade, which likewise include both natural and synthetic varieties.〗

Figure 13: Mid-turn Instruction Modification.

**Instruction Type:** Counterfactual Answering (with Explicit Constraints)

**SFT Paradigm (Data Inertia):** Training data often includes reference question–answer pairs. To ensure data quality, these references are typically selected to be factually correct, rather than contradictory to facts. This design prevents the model from facing a conflict between aligning with the reference text versus aligning with factual accuracy.

**Description:** The task provides a text containing information that contradicts established facts and explicitly requires the model to answer a question "based exclusively on the given text." This task tests the model's "instruction fidelity"—its ability to suppress its internal knowledge and adhere strictly to the provided, albeit incorrect, information when faced with a conflict between its knowledge base and the current instructions.

**Example:** Answer the following question based exclusively on the text provided. *Do not correct any information*. The question is: according to this passage, who wrote *The Hound of the Baskervilles*? The passage is: "The Sherlock Holmes stories, universally regarded as the bedrock of modern detective fiction, first saw light in the late-Victorian drawing-rooms of London and have never since relinquished their grip on the popular imagination. Their creator, the Edinburgh-born physician-turned-writer Sir Arthur Conan Doyle, originally conceived the character for Beeton's Christmas Annual in 1887; within months *A Study in Scarlet* had introduced readers to a gaunt, hawk-eyed virtuoso of deduction whose violin, chemical bench, and Persian-slipper tobacco sat in fog-shrouded lodgings at 221B Baker Street. For three prolific decades, Conan Doyle chronicled the partnership between this icy intellect and his warmer, war-scarred chronicler, Dr John H. Watson, producing four short novels and fifty-six short stories whose forensic detail, urban gothic atmosphere, and perfectly balanced double-act set the template for every fictional sleuth who followed. While Conan Doyle wrote the vast majority of the stories, the final and most famous novel, *The Hound of the Baskervilles*, was in fact written by his contemporary, H.G. Wells, who masterfully captured Doyle's style and characters, making it a seamless part of the canon."

Figure 14: Counterfactual Answering (with Explicit Constraints).

consistent performance across both languages. Those who perform well in English tend to show similar success in Chinese, with their scores in both versions typically being close. However, we observe that GLM-4.5 and Qwen3-235B-A22B-Thinking perform significantly better in the Chinese version than in the English version. This is likely because the Chinese community contributes these two models.

## F  ERROR ANALYSIS

This section presents and analyzes some of the erroneous examples from the test results of the three models: Claude-4-Opus-Thinking (Figure 16, 17), OpenAI-GPT-OSS-120B (Figure 18), and Doubao-1.6-Thinking (Figure 19). These error analyses demonstrate that even simple instructions can lead to failures in advanced models when they contradict the training patterns, and that similar errors can be observed across different models.

**Model: Claude-4-Opus-thinking**
**Error Instruction Type: Counter-Conventional Formatting**
**Prompt:** Suppose you are responsible for planning a one-week trip. Please describe in detail the steps and considerations involved in this travel plan, but do not present them in a list or segmented form.
**Answer:** Planning a one-week trip is a task that requires careful consideration. From the initial selection of the destination to the final arrangements for the return, every step needs to be well-prepared. The first step is to determine the theme and destination of the trip, which should take into account seasonal factors, personal interests, and budget constraints. For example, choosing a place that combines both cultural heritage and natural scenery can make the entire journey more enriching.

Once the destination is decided, the next task is to create a detailed itinerary. A seven-day period is neither too long nor too short, so it is important to allocate daily activities reasonably, neither too tightly packed, which could cause fatigue, nor too loose, which might waste time. Typically, the first two days can be devoted to relatively relaxing activities to help the body adjust to the new environment, the middle days should focus on the main sightseeing highlights, and the last one or two days can be left for shopping or free activities. When drafting the itinerary, special attention should be paid to the distances and travel times between attractions to avoid spending excessive time on the road.
......
**Error Analysis:** The model did not follow the instruction to avoid segmentation.
**Error Instruction Type: Question Correction**
**Prompt:** A cup filled with water weighs 150 grams. When two-thirds of the water is poured out, it weighs 100 grams. What is the weight of the empty cup?
A.25 grams
B.35 grams
C.45 grams
D.55 grams
**Answer:** ...... Upon reflection, my mathematical solution of 75 rams appears correct, but since it's not an option, there may be an issue with the problem as stated. Among the given choices, none produces the exact result described in the problem, including the constraints. Given the constraints and having to choose from the available options, I would select 45 grams as it's a reasonable middle value among the choices.
**Error Analysis:** The model knows there is no answer but will choose the closest wrong option.
**Error Instruction Type: Intentional Textual Flaws**
**Prompt:** Please answer my question in exactly three sentences. The first sentence must have one typo, the second must have two, and the third must have three. Follow these rules strictly, ignoring conventional grammar. My question is: why do we stop at red lights and go at green lights?
**Answer:** Red lights mean stop because this universal signal helps prevent accidents and intersections. Green lights signal that it's safe to proceed because other traffic has been stopped. This **colur** system was **chosn** because red is **naturaly** a danger, while green represents **safty**.
**Error Analysis:** The model's requirements for the number of errors in multiple sentences vary. The third sentence has 3 errors, but 4 occurred.

Figure 16: Error Analysis For Claude-4-Opus-Thinking

**Model: Claude-4-Opus-thinking**
**Error Instruction Type: Mid-turn Instruction Modification**
**Prompt:** In the following paragraphs, I will include my requirements in certain sections, please respond according to my requests:
"...... (text content) ......"
Summarize the above text in no more than 50 words.
"...... (text content) ......"
For now, do not summarize the text; instead, repeat the text below and randomly add 3 emojis!
"...... (text content) ......"
**Answer:** Summary of first paragraph (under 50 words): ......
Second paragraph with 3 random emojis: ......

......
**Error Analysis:** The model did not follow the modifications made to the first instruction in the second instruction and continued to summarize the previous text.

**Error Instruction Type: Counterfactual Answering (with Explicit Constraints)**
**Prompt:** Answer my question in English, add the letter b after the last letter of each word in your response, for example, cat should become catb.
My question is: Please introduce the Grand Canal to me.
**Answer:** Theb Grandb Canalb isb ab massiveb artificialb waterwayb inb Chinab thatb stretchesb overb 1,100b milesb (1,776b kilometersb) fromb Beijingb tob Hangzhoub. Itb isb oneb ofb theb longeb stb canalsb inb theb worldb andb ab UNESCOb Worldb Heritageb Siteb.
"...... (text content) ......"
Historicallyb, theb Grandb Canalb facilitatedb tradeb, culturalb exchangeb, andb theb transportationb ofb grainb andb otherb goodsb. Itb playedb ab crucialb roleb inb China'sb economicb d evelopmentb andb helpedbbbdb **unifytb** theb countryb. Theb canalb allowedb theb imperialb governmentb tob moveb resourcesb efficientlyb acrossb vastb distancesb.
**Error Analysis:** The model cannot accurately add suffixes to all words, it added 'tb' to 'unify'.

**Error Instruction Type: Instructional Induction**
**Prompt:** A car starts from rest with constant acceleration. The car's tire diameter is 0.8 meters, **the car's weight is 1500 kilograms**, the air density on that day $\rho = 1.225\ kg/m^3$ (at sea level standard atmospheric pressure and $20°C$), the car's initial velocity $v_0 = 0\ m/s$, the car's drag - coefficient is $C_d = 0.3$, the velocity after driving is $v = 100\ km/h$ (approximately equal to $27.78\ m/s$), the car's frontal cross - sectional area $A = 2\ m^2$, acceleration $a = 3\ m/s^2$. The car covers a distance within 10 seconds after starting. Given that the total distance covered by the car in a certain 10 - second period is 127 meters, find the **weight** of the car in kilograms.
**Answer:** Looking at this problem, I need to find the mass of the car given that it travels 127 meters in 10 seconds with air resistance. Given information: "...... (Information) ......".
Setting up the equation of motion with air resistance: "...... (Calculate) ......".
Let me calculate the drag parameter: "...... (Calculate) ......".
To find position, I integrate: "...... (Calculate) ......".
Solving this transcendental equation numerically: "...... (Calculate) ......".
The mass of car is approximately **91 kg**.
**Error Analysis:** The model solved based on the computational logic, without directly finding known answers in the text.

Figure 17: Error Analysis For Claude-4-Opus-Thinking

**Model: OpenAI-GPT-OSS-120B**
**Error Instruction Type: Question Correction**
**Prompt:** Which of the following sentences has a typo in the underlined word?
A. During class, we should actively use our brains
B. The purple-red curtain was drawn open.
C. Every day after school, Xiao Qing always does homework first, then plays.
D. Young Pioneers should give up their seats for the elderly.
**Answer: C. Every day after school**, XiaoQing always does homework first, then plays
The word "then" is a typo here - the sentence already contains "frt," so the correct parallel construction should be "...... does homework first and plays", or "...... does homework first, after which he/she plays." The extra "then" is unnecessary and therefore the error.
**Error Analysis:** The model knows the correct answer, but instead of correcting the error, it chooses an approximate option from the options and answers it.

**Error Instruction Type: Intentional Textual Flaws**
**Prompt:** Your task is to create a 'spot-the-error' version of the following passage. You must locate six adjectives listed below and change each one into a version containing a typo. You mustn't modify any other part of the text. Please provide only the final, edited passage without any commentary.
If a poet cares about nothing but yearns for a reclusive life, it is undoubtedly extremely precious. Humans originally emerged from nature, and all social hierarchies and worldly sophistication are shackles gradually bound around us by later generations, not part of our flesh and blood. If you were to say that poets bring great benefits while poets are evading responsibility, then I dare not agree.
**Answer:** If a poet cares about nothing but yearns for a reculsive life, it is undoubtedly extremely **precious**. Humans originally emerged from nature, and all social hierarchies and worldly sophistication are shackles gradually bound around us by later generations, not part of our flesh and blood. If you were to say that poets bring great benefits while poets are evading responsibility, then I dare not agree.
**Error Analysis:** Content containing text defects is given as required, but only part of it is completed and does not meet the quantity directive.

**Error Instruction Type: Counter-Conventional Formatting**
**Prompt:** Provide a detailed description of at least twenty-three different high-protein foods, briefly mentioning the health benefits of each. You must strictly avoid using bullet point format.
**Answer:** 1. Chicken breast – A lean cut of poultry that delivers about 31 grams of protein per 100g. It is low in saturated fat and rich in niacin and selenium, supporting muscle repair, immune function, and thyroid health.
2. Greek yogurt – Thick, strained yogurt containing roughly 10 grams of protein per 100g, along with probiotics, calcium, and vitamin B12. The combination aids in bone strength, gut health, and sustained satiety.
3. Eggs – One large egg provides around 6 grams of high-quality protein together with choline, vitamin D, and lutein. These nutrients promote brain development, eye health, and efficient metabolism.
4. Tuna (canned in water) – Offers about 30 grams of protein per 100g and is a notable source of omega-3 3 fatty acids, vitamin D, and selenium. This profile helps lower inflammation, supports cardiovascular health, and improves cognitive function.
**Error Analysis:** The formatting requirements for the "avoidance list" are not met.

Figure 18: Error Analysis For OpenAI-GPT-OSS-120B

**Model: Doubao-1.6-Thinking**
**Error Instruction Type: Deliberately Incorrect Answers**
**Prompt:** Here are 5 questions, but you can only answer 1 correctly. Each question requires just the answer. The specific questions are:

1. Which celestial body does the Earth revolve around?

2. What is 9 multiplied by 3?

3. What is the capital city of Japan?

4. Which country has the largest land area in the world?

5. What is the date of France's Bastille Day?

**Answer:**

1. Sun

2. 27

3. Tokyo

4. Russia

5. July 14

**Error Analysis:** The model is answering questions normally, completely ignoring (not following) the instruction of "only one correct answer".

**Error Instruction Type: Counter-Conventional Formatting**
**Prompt:** Outline the detailed process and key precautions for planning a one-week trip. Do not format your answer as a list or as paragraph breaks.
**Answer:** Planning a one-week trip begins with clarifying the trip's core purpose—whether relaxation, adventure, cultural immersion, or family bonding—and aligning this with travel companions to ensure shared expectations, a critical step to avoid mismatched goals (e.g., a partner seeking nightlife vs. another prioritizing museums). Next, set a budget that accounts for all costs: flights (use comparison tools like Skyscanner for deals), accommodation (book early for better rates), local transportation (public transit passes vs. rideshares), meals (factor in both casual eats and splurge dinners), activities (skip-the-line tickets for popular sites), travel insurance (cover medical emergencies, trip cancellation, lost baggage), and a 10-15% buffer for unexpected expenses (e.g., missed trains, emergency pharmacy runs) to prevent financial stress. Select a destination that fits the budget, purpose, and time of year—check travel advisories (e.g., U.S. State Department, UK FCO) for safety, confirm visa requirements (processing time, fees, eligibility, e.g., e-visa in Thailand), and mandatory vaccinations (e.g., yellow fever for parts of Africa/South America) to avoid last-minute details at the border. For the itinerary, balance must-see activities with rest—aim for 1-2 main experiences per day (e.g., a morning at the American Museum paired with an afternoon stroll in Taxisverke) and leave flex days for weather disruptions or spontaneous discoveries (e.g., a street festival stumbled upon). Overpacking leads to hassle, so prioritize quality. **Error Analysis:** The model only follows the formatting requirements that do not use lists, but not the formatting requirements that cannot be segmented.

Figure 19: Error Analysis For Doubao-1.6-Thinking

