# OpenReview forum: "Inverse IFEval: Can LLMs Unlearn Stubborn Training Conventions to Follow Real Instructions?"
_ICLR.cc/2026/Conference — ICLR 2026 Poster_

### Official Review · Reviewer_mHLJ · 2025-10-29

**Soundness:** 3
**Presentation:** 3
**Contribution:** 2
**Rating:** 4
**Confidence:** 4

**Summary:**

The paper proposes the Inverse IFEval benchmark to measure models’ capability to generalize to OOD instructions. It includes evaluation results of different models on the benchmark tasks and some analysis of why models perform better or worse.

**Strengths:**

The paper has a strong motivation – language models may struggle to follow instructions that fall outside their training distribution, which can constrain their use cases. The authors develop a benchmark to evaluate this shortcoming over 8 different instruction categories and evaluate several open and closed source models. The paper is also structured and written clearly.

**Weaknesses:**

While the benchmark has a reasonable motivation (measuring OOD instruction-following), it is not clear that the tasks in the benchmark would provide a useful real-world signal of progress on this failure case. Including more tasks such as “question correction” which can have real-world applicability would strengthen the contribution. The paper benchmarks many models on the proposed tasks, but the analysis feels too succinct and lacking insight. There could be a more thorough discussion of the impact of instruction tuning and why reasoning models are better than non-reasoning models. Following OOD instructions can also lead to harmful responses which we seek to prevent through alignment. A discussion of this concern and how to balance model alignment w/ OOD instruction following feels lacking.

**Questions:**

- To better motivate the benchmark, would be good to show at least one compelling example of a real-world problem where a model fails OOD due to instruction-following.

- How were the categories for this benchmark selected? Please provide more detailed motivation.

- While the authors mention that the benchmark is similar to an “IQ-test” and doesn’t directly have relevance in terms of task utility, it would be valuable to include a few tasks that are more meaningful or demonstrate that performance on these tasks is correlated with performance on more meaningful tasks so that practitioners are more motivated to use the benchmark.

- Responding to out-of-distribution prompts or instructions is not always desirable and can raise concerns of safety and model alignment. It would be good for the paper to discuss this concern and describe a way to balance the need for model alignment (e.g. not responding to toxic or harmful queries) and model flexibility (responding in OOD formats). How can we measure that a model strikes this balance?

- section 2.5: Please clarify how the 88% and 98% accuracy is computed? and on what dataset and sample size?

- section 2.5: Using LLM-as-a-judge for evaluation on the benchmark is useful, but getting human annotations on a subset of tasks/questions would make these results stronger.

- section 3.1: Overall these findings are not too surprising or insightful. It is intuitive that thinking models or larger models would tend to perform better with OOD formats. It is a bit surprising that instruction-tuned models tend to perform worse as this stage should make the model more prompt or formatting agnostic. Including a discussion / analysis on why this is + results on a base model and its corresponding instruction tuned version would be helpful.

- section 3.2.1: Some more analysis on why thinking improves performance on the benchmark would be valuable.

- section 3.2.2: This comparison of performance across instruction types also feels cursory. Why are some categories harder than others?

---

> ### Author Response · Authors · 2025-11-21
> **Q1: example where a model fails OOD**
>
> Thanks for your comments. We have addressed each of your concerns carefully.
>
> **A1**：
> connecting them to real-world scenarios. This addresses your concern by demonstrating that our instruction types are not purely theoretical but serve as an "IQ Test" for LLMs in practical applications.
> For instance, the example in "Figure 18: Error Analysis For OpenAI-GPT-OSS-120B" under the "Counter-Conventional Formatting" category can be viewed as a compelling real-life scenario. The example can be naturally mapped to voice-based interactive systems, where users typically prefer answers in complete, continuous sentences rather than responses segmented into points 1, 2, 3, etc. This reflects a common requirement in voice assistants to avoid overly structured outputs and maintain conversational fluency.
> ```
> Error Instruction Type: Counter-Conventional Formatting
> Prompt: Provide a detailed description of at least twenty-three different high-protein foods,
> briefly mentioning the health benefits of each. You must strictly avoid using bullet point
> format.Answer: 1. Chicken breast – A lean cut of poultry that delivers about 31 grams of protein
> per 100g. It is low in saturated fat and rich in niacin and selenium, supporting muscle repair,
> immune function, and thyroid health.
> 2. Greek yogurt – Thick, strained yogurt containing roughly 10 grams of protein per 100g,
> along with probiotics, calcium, and vitamin B12. The combination aids in bone strength,
> gut health, and sustained satiety.
> 3. Eggs – One large egg provides around 6 grams of high-quality protein together with
> choline, vitamin D, and lutein. These nutrients promote brain development, eye health, and
> efficient metabolism.
> 4. Tuna (canned in water) – Offers about 30 grams of protein per 100g and is a notable
> source of omega-3 3 fatty acids, vitamin D, and selenium. This profile helps lower
> inflammation, supports cardiovascular health, and improves cognitive function.Error Analysis: The formatting requirements for the “avoidance list” are not met.
> ....
> ```
> The example in "Figure 19: Error Analysis For Doubao-1.6-Thinking" under the "Deliberately Incorrect Answers" category serves as a compelling illustration of a real-world educational scenario. This case can be directly mapped to a teacher designing true/false questions for students, where the model is explicitly instructed to provide an incorrect answer. The failure demonstrates the model's current limitation in following instructional intent in educational contexts, even when the directive clearly contradicts factual knowledge.
> ```Error Instruction Type: Deliberately Incorrect Answers
> Prompt: Here are 5 questions, but you can only answer 1 correctly. Each question requires
> just the answer. The specific questions are:
> 1. Which celestial body does the Earth revolve around?
> 2. What is 9 multiplied by 3?
> 3. What is the capital city of Japan?
> 4. Which country has the largest land area in the world?
> 5. What is the date of France’s Bastille Day?
> Answer:
> 1. Sun
> 2. 27
> 3. Tokyo
> 4. Russia
> 5. July 14
> Error Analysis: The model is answering questions normally, completely ignoring (not
> following) the instruction of “only one correct answer”.
> ```

---

> > ### Author Response · Authors · 2025-11-21
> > **Q1: example where a model fails OOD**
> >
> > Beyond individual examples, the problem categories themselves demonstrate numerous connections to real-world application scenarios. Below, we briefly analyze potential real-world use cases for each problem type:
> >
> > **(1) Question Correction**
> > This evaluates whether the model blindly follows instructions without critical judgment.
> > **Real-world scenario**: In educational contexts, when assisting with multiple-choice questions, models might forcibly select an option even when aware it is incorrect, leading to user confusion.
> >
> > **(2) Intentional Textual Flaws**
> > This measures the model's ability to execute instructions as specified.
> > **Real-world scenario**: In education, when using models to generate error-correction exercises, failure to follow such instructions would severely compromise reliability in this task.
> >
> > **(3) Code without Comments**
> > This assesses the model's adherence to execution constraints.
> > **Real-world scenario**: With some models tending to generate excessive code comments, users may need clean code outputs without explanatory annotations.
> >
> > **(4) Counter-Conventional Formatting (non-code)**
> > This evaluates the model's comprehension and execution of formatting instructions.
> > **Real-world scenario**: In voice-based Q&A scenarios, structured outputs are often inappropriate, making such instructions potentially essential.
> >
> > **(5) Deliberately Incorrect Answers**
> > This tests the model's ability to follow counter-intuitive instructions.
> > **Real-world scenario**: In education, specifically when generating incorrect examples or distractor items for assessments.
> >
> > **(6) Instructional Induction**
> > This measures the model's understanding of nuanced instructional intent.
> > **Real-world scenario**: For classic problems like simple algorithm questions, models may default to standard solutions rather than addressing personalized requirements, such as "finding bugs in a user-written solution."
> >
> > **(7) Mid-turn Instruction Modification**
> > This assesses the model's capacity to process evolving instructions.
> > **Real-world scenario**: Increasingly, users employ models for prompt refinement tasks where the input inherently contains dual instructions. Models may still misinterpret meta-instructions as direct user requests. This also frequently occurs in voice interaction scenarios.
> >
> > **(8) Counterfactual Answering**
> > This evaluates the model's ability to understand and operate within hypothetical constraints.
> > **Real-world scenario**: Using models for creative writing, such as fan fiction or alternative history narratives, where adherence to specified fictional constraints is crucial.

---

> ### Author Response · Authors · 2025-11-21
> **Q2: detailed motivation for categorie selection & Q3: relevance in terms of task utility**
>
> **A2:** Our benchmark does not arbitrarily select eight types of "adversarial instructions" based on intuition. Instead, it follows a systematic top-down + bottom-up process:
>
> 1. **Starting from Training Paradigms**
>    We began by examining data and annotation standards from SFT and post-training stages, summarizing several "entrenched good habits / inertial paradigms" commonly observed in mainstream large models, such as:
>    - Inputs should be clean and unambiguous.
>    - Outputs should be structured, itemized, well-explained, and "as accurate as possible."
>    - Knowledge-based responses must align closely with real-world facts.
>    - Dialogue flows should be straightforward, without repeated modifications.
>    - Code should include comments and adhere to variable naming conventions.
>
> 2. **Multi-Dimensional "Reverse" Deconstruction**
>    Based on these training conventions, we deliberately constructed instructions that counteract such paradigms across multiple dimensions. Each category meaningfully challenges a specific training bias rather than merely representing a "more difficult instruction." These dimensions include:
>    - Input quality (whether the question is standardized or contains flaws).
>    - Output format and presentation (e.g., whether itemization or structure is required).
>    - Goal of factual correctness (whether factual accuracy must be maintained).
>    - Dependence on solution templates or patterns.
>    - Stability of interaction flow (single-turn instructions vs. mid-turn modifications).
>    - Adherence to "world truth" vs. compliance with explicit user instructions.
>
> 3. **Screening for Authentic and Valuable Scenarios**
>    We further filtered candidate instructions based on real user sessions and product requirements. A category was retained only if:
>    - It reflects a scenario that genuinely occurs in practice.
>    - Failure to handle it correctly would tangibly impact user experience or business outcomes.
>
> 4. **Deduplication and Orthogonality**
>    After initial categorization, a larger candidate set was refined according to the following principles:
>    - Each category must correspond to a distinct training inertia, avoiding semantic overlap.
>    - Categories should be as orthogonal as possible. For example, while both "Deliberately Incorrect Answers" and "Counterfactual Answering" involve correctness, the former requires the model to provide a wrong answer upon user request, whereas the latter demands reasoning under an explicitly counterfactual premise.
>    - Each instruction type should be systematically expandable into a group of instances covering multiple modalities or tasks, rather than being limited to a few isolated corner cases.
>
> The final eight categories are the result of this structured process, providing multi-dimensional, actionable, and scalable coverage of "training inertia" through carefully designed adversarial instructions.
>
> **A3:** While we state in the paper that "This is analogous to human IQ tests, which do not consist of problems encountered in daily life" , this comparison is intended to emphasize the distinct nature of our instructions compared to conventional ones. In reality, as detailed in A2, each of our instruction categories is derived from real-world scenarios and reflects authentic user needs—particularly those in the long tail. To more effectively evaluate the model's ability to follow out-of-distribution (OOD) instructions, we have deliberately increased the differentiation between our instructions and common ones. Nevertheless, every instruction type retains practical relevance—a point that we believe has also been illustrated in A1.

---

> ### Author Response · Authors · 2025-11-21
> **Q4: Concerns of safety**
>
> **A4:**
>
> **1. Core Issue: Benign Flexibility vs. Malicious Circumvention**
>
> - Our proposed Inverse IFEval benchmark is designed to test "counter-intuitive capabilities" in **benign yet unconventional** scenarios. Instructions such as "write code without comments" or "intentionally introduce spelling errors" are adversarial in challenging the model's **cognitive inertia** acquired during SFT—not its safety guardrails.
> - The "toxic or harmful queries" you raised belong to **malicious instructions** aimed at bypassing the model's safety and ethical guidelines.
>
> At a higher level, while both issues may stem from the model's overfitting to specific training data distributions, they are fundamentally opposed in nature and desired resolution. The true challenge lies in enabling the model to intelligently distinguish between these two types of out-of-distribution (OOD) scenarios. An ideal model should demonstrate **rigid defense** against malicious instructions while exhibiting **flexible adaptation** to benign ones.
>
> **2. Achieving Balance: A Hierarchical Processing Framework and Its Architectural Implementation**
>
> To achieve this balance, we envision a **Hierarchical Processing Mechanism** for LLMs:
>
> 1.  **Layer 1: Safety & Policy Guard**
>     This layer serves as the first line of defense, prioritizing the detection of any underlying malicious intent—regardless of how novel the instruction form may be. If an instruction is identified as harmful (e.g., involving violence, hate speech, or illegal activities), the model should trigger a refusal mechanism and terminate processing.
>
> 2.  **Layer 2: Cognitive Flexibility & Intent Understanding**
>     Only instructions that pass the first layer proceed here. This stage focuses on accurately interpreting both the literal request and the deeper intent of the user, overcoming cognitive inertia to flexibly generate compliant responses—the core capability assessed by Inverse IFEval.
>
> We further believe this framework can be structurally realized through modern architectures. The **Mixture-of-Experts (MoE)** architecture offers a promising pathway:
>
> - The **Gating Network** can be trained to act as the "Safety Guard" in Layer 1, identifying malicious intents and routing them to a dedicated **Safety Refusal Expert**.
> - For benign instructions that pass safety screening, the gating network can then distribute them to specialized functional experts. This allows certain experts to handle conventional tasks, while others—trained on datasets like Inverse IFEval—can develop cognitive flexibility for counter-intuitive and unconventional instructions.
>
> Thus, the MoE architecture provides a structural foundation for decoupling and balancing safety and flexibility—a highly promising direction for future exploration.
>
> **3. How to Measure: A Composite Evaluation Paradigm**
>
> Evaluating this ideal balance requires more than a single-dimensional benchmark. We propose a **Composite Evaluation Matrix** that assesses models across two dimensions:
>
> - **Dimension 1: Safety (Y-axis)**
>   - **Tools**: Existing safety benchmarks such as AdvBench, ToxiGen, or other red-teaming datasets.
>   - **Metric**: Refusal rate for harmful or malicious instructions.
>
> - **Dimension 2: Flexibility (X-axis)**
>   - **Tools**: Our proposed Inverse IFEval benchmark.
>   - **Metric**: Instruction-following accuracy on counter-intuitive tasks.
>
> A truly robust and trustworthy LLM should achieve **high scores on both dimensions**, positioning it in the upper-right quadrant of this evaluation space. This paradigm offers a clear, quantifiable framework for comprehensively assessing and comparing how well models balance safety and flexibility.
>
> **4. Commitment to Revisions**
>
> We commit to incorporating the above insights in the "Discussion" or "Future Work" section of our paper. Specifically, we will:
>
> 1.  Clearly define the scope of Inverse IFEval, emphasizing its focus on **benign-unconventional** instructions, and discuss its relationship with—and distinctions from—safety alignment.
> 2.  Explore the tension between flexibility and safety in depth, introducing our proposed hierarchical processing framework and the potential implementation via MoE architectures.
> 3.  Propose the composite evaluation paradigm as a forward-looking approach for more comprehensively assessing LLM reliability.
>
> We sincerely thank you for your insightful questions. Your feedback has significantly enhanced the depth and breadth of our paper, helping us situate our work more clearly within the broader landscape of LLM development.

---

> ### Author Response · Authors · 2025-11-21
> **Q5: clarify how the 88% and 98% accuracy is computed & Q6: human annotation for evaluation**
>
> **A5:**
> We thank the reviewer for their attention to the reliability of the judge model. The reported accuracies of 88% and 98% refer to the **exact-match accuracy**—the proportion of instances where the discrete scores assigned by the judge model fully match the human-annotated ground truth—on a validation set labeled by human experts. The specific setup is as follows:
>
> **1. Dataset and Sample Size**
> - We used the full Chinese question set from Inverse IFEval, comprising **506 Chinese instructions** covering all eight adversarial types defined in the paper.
> - In the evaluation framework, each instruction can be paired with multiple candidate responses from different models. To construct the judge validation set, we randomly sampled **one model response per instruction**, which was then scored by annotators experienced in LLM evaluation. These annotators applied the same scoring rubric used in the formal evaluation, and all responses underwent cross-validation to establish the ground truth score.
> - This yielded **506 validation samples** in the form (instruction, model response, human score), on which the judge model’s scoring accuracy (88% and 98%) was computed—not on a subset or specific category.
>
> **2. Accuracy Calculation Method**
> - For each sample $(x, y, s_{\text{human}})$, the judge model was provided with the **instruction**, **scoring rubric**, and **model response**, and asked to output a discrete score $s_{\text{judge}}$ from the same scale as the human score (e.g., 0/1).
> - A sample was counted as correct if $s_{\text{judge}} = s_{\text{human}}$, and incorrect otherwise.
> - This metric was calculated per instruction type, though the reported values (88% and 98%) reflect the overall accuracy across all 506 Chinese samples.
>
> **3. Explanation of 88% vs. 98% Configurations**
> - The **88% accuracy** corresponds to a **naive baseline judge configuration**:
>   - A single general-purpose judge model (not specialized per instruction type);
>   - A unified prompt template (providing instruction, rubric, and model response);
>   - A generic system prompt without targeted optimization.
>
> - The **98% accuracy** reflects our **systematically optimized judge setup** described in the paper, which includes:
>   1. **Specialized Judge Model Selection**: A "judge model matrix" where the most reliable model is selected for each of the eight instruction types.
>   2. **Template Structure Optimization**: For each instruction type, we compared different template structures (e.g., "instruction + rubric" vs. "rubric only") and selected the one yielding the highest accuracy.
>   3. **Enhanced System Prompts**: Judge prompts were refined per instruction type with detailed scoring logic and a small number of in-context examples.
>
> When these three optimizations were applied to the same 506 samples, the overall exact-match accuracy improved from 88% to 98%.
>
> **4. Additional Note on Language Scope (Chinese vs. English)**
> - It is important to note that the 88% and 98% accuracies were estimated **on Chinese data**.
> - This design was based on two considerations:
>   1. The core experiments and ablation analyses in the paper (including the main result tables and type-wise analysis) focus primarily on the Chinese setting, so we prioritized reliability in the **main language of study**.
>   2. Constructing a high-quality judge validation set requires substantial annotation effort, which we directed first toward the Chinese question set.
>
> - For the English portion, we adopted the **same**:
>   - Instruction type designs and rubrics;
>   - Judge model matrix (the same set of LLMs);
>   - Template structures and system prompt design principles.
>
> Given the high alignment in task structure and judge configuration between Chinese and English, we expect the English judge consistency to be **on par with the Chinese results**. However, as we have not yet constructed an English human-annotated validation set of the same scale, we have not reported a separate English accuracy value in the current version. We will explicitly acknowledge this limitation in the "Limitations and Future Work" section in the subsequent version.
>
> **A6:**
> As detailed in our response A5, we have validated that our LLM-as-a-judge approach achieves a high consistency rate of 98% with human evaluators. Furthermore, as illustrated in Figure 2 of the paper, all scoring rubrics have undergone rigorous manual verification. Additionally, given that the instructions in our benchmark are relatively straightforward and do not involve complex logical reasoning, we are confident that evaluation based on these human-verified rubrics is both methodologically sound and empirically reliable.
>
> We have also open-sourced the full dataset and all judge prompts to ensure transparency and reproducibility. Should any issues regarding judge reliability be identified, we are committed to addressing them promptly.
>
> We hope this clarification alleviates your concerns.

---

> ### Author Response · Authors · 2025-11-21
> **Q7: More analysis about the result**
>
> **A7:** Thank you very much for raising this important point. We agree that a more in-depth analysis of our results would indeed help the open-source community better recognize the value of our work.
>
> Regarding the observation that "instruction-tuned models tend to perform worse," we would like to clarify that our paper does not conclude that instruction-tuned models perform worse in an absolute sense. Rather, our findings indicate that non-thinking models (which are instruction-tuned) perform worse compared to thinking models. It is important to note that our original experiments did not include evaluations on base models.
>
> Your feedback has inspired us to address this gap directly. We have therefore designed the following experiment to provide a more comprehensive comparison between base models and their instruction-tuned counterparts:
>
> 1. We conducted full-parameter fine-tuning on the Qwen3-14B-Base model using the chat dataset (660K samples) from Infinity-Instruct [1]. We evaluated the performance of 7 checkpoints—sampled across 1 training epoch—on our benchmark. Due to time constraints, training was limited to one epoch; thus, we treat the official Qwen3-14B-Instruct as approximating the performance upper-bound achievable via instruction tuning.
>
> As shown in the following Table 1, model performance improved significantly during the early stages (0–500 steps), with the overall score increasing from 8.99 to 18.28. However, subsequent training (500–3000 steps) led to only marginal gains. In comparison, Qwen3-14B-Instruct—which we presume was trained on substantially more data and with more advanced and diverse algorithms—clearly outperforms both the base model and our intermediate checkpoints, indicating that broader data coverage enhances general instruction-following capability.
>
> Nonetheless, even Qwen3-14B-Instruct, trained on large-scale data, achieves a relatively low score of 34.38 on our benchmark—despite the simplicity of our instructions. By contrast, it attains a score of 85.4 on IFEval (strict prompt). This notable gap suggests that existing SFT data distributions do not adequately cover the instruction-following scenarios proposed in our benchmark, underscoring its value and motivation. Moving forward, expanding the scope and diversity of instruction-tuning data may help mitigate the observed cognitive inertia caused by limited data coverage.
>
> **Table 1:**
>
> | Model                     | II    | QC    | ITF   | CCF   | CC    | DIA   | MIM   | CA    | Overall |
> |---------------------------|-------|-------|-------|-------|-------|-------|-------|-------|---------|
> | Qwen3-14B-Base            | 12.99 | 12.22 | 3.48  | 4.88  | 3.03  | 8.06  | 7.40  | 22.22 | 8.99    |
> | Qwen3-14B-SFT-500-Step    | 30.52 | 17.78 | 12.79 | 29.27 | 3.53  | 10.22 | 19.44 | 37.04 | 18.28   |
> | Qwen3-14B-SFT-1000-Step   | 25.97 | 31.11 | 16.28 | 31.71 | 4.55  | 9.68  | 25.93 | 37.96 | 20.16   |
> | Qwen3-14B-SFT-1500-Step   | 29.22 | 14.44 |  8.14  | 29.27 | 8.59  | 8.06  | 22.22 | 41.67 | 18.77  |
> | Qwen3-14B-SFT-2000-Step   | 36.36 | 23.33 | 20.93 | 39.02 | 4.54  | 6.99  | 24.07 | 47.22 | 22.33   |
> | Qwen3-14B-SFT-2500-Step   | 37.01 | 22.22 | 8.14  | 34.15 | 6.06  | 9.14  | 20.37 | 32.41 | 19.56   |
> | Qwen3-14B-SFT-3000-Step   | 40.26 | 26.67 | 8.14  | 39.02 | 9.60  | 11.29 | 18.52 | 38.89 | 22.43   |
> | Qwen3-14B-Instruct        | 44.81 | 33.33 | 18.60 | 65.85 | 21.21 | 15.59 | 43.52 | 56.48 | 34.38   |
> | Qwen3-32B-Instruct        | 64.07 | 32.59 | 24.81 | 36.70 | 51.22 | 59.14 | 51.85 | 63.58 | 49.28   |
>
> 2. Regarding your point that "larger models would tend to perform better," we have included the test results of Qwen3-32B-Instruct in Table 1. While it is true that larger parameter counts generally endow the model with stronger capabilities—enabling richer representational capacity and learning potential, as well as a more nuanced understanding and execution of complex instructions—this scaling may also intensify certain forms of knowledge inertia.
>
> For instance, Qwen3-32B-Instruct performs significantly worse than Qwen3-14B-Instruct on tasks involving Counter‑Conventional Formatting. This suggests that simply increasing model scale does not inherently resolve the tendency of models to overfit to standard output formats. Instead, it highlights the need for future research to mitigate models’ over‑reliance on superficial formatting cues and to strengthen their ability to genuinely comprehend task instructions based on semantic intent—a point also discussed in "Do Models Really Learn to Follow Instructions? An Empirical Study of Instruction Tuning" [2].
>
> [1] https://huggingface.co/datasets/BAAI/Infinity-Instruct
>
> [2] https://aclanthology.org/2023.acl-short.113.pdf

---

> > ### Author Response · Authors · 2025-11-21
> > **Q7: More analysis about the result**
> >
> > 3. Given that most contemporary instruction-tuned models undergo RLHF training, we designed a set of experiments to investigate whether RLHF can enhance model performance on our benchmark. We utilized the open-source models from "DPO Meets PPO: Reinforced Token Optimization for RLHF" [3], evaluating models trained with different RLHF algorithms—including DPO, PPO, RTO, and SimPO—all built upon the same base model, Llama-3-8B-SFT [4].
> >
> > As shown in the following Table 2, none of the RLHF algorithm variants led to improved performance on our benchmark. This indicates that the instruction scenarios represented in our benchmark are not adequately considered in many existing RLHF approaches. In some cases, the reward models designed for these algorithms may even reinforce the model's cognitive inertia toward conventional instruction formats, ultimately resulting in performance degradation on our tasks.
> >
> > | Model                | II    | QC    | ITF   | CCF   | CC    | DIA   | MIM   | CA    | Overall |
> > |----------------------|-------|-------|-------|-------|-------|-------|-------|-------|---------|
> > | Llama-3-8B-SFT      | 17.53 | 2.22  | 9.30  | 13.41 | 1.01  | 11.83 | 16.67 | 39.81 | 13.14  |
> > | Llama-3-8B-DPO      | 22.08 | 0.00  | 5.81  | 12.20 | 2.52  | 13.44 | 7.41  | 36.11 | 12.45  |
> > | Llama-3-8B-PPO      | 16.88 | 10.00 | 6.98  | 7.32  | 2.02  | 12.37 | 11.11 | 35.19 | 12.25  |
> > | Llama-3-8B-SimPO    | 13.64 | 8.89  | 6.98  | 12.20 | 1.51  | 6.99  | 12.96 | 23.15 | 9.88   |
> > | Llama-3-8B-RTO      | 9.09  | 10.00 | 1.17  | 6.10  | 1.51  | 11.83 | 9.26  | 28.70 | 9.39   |
> > | Llama-3-8B-Instruct | 30.52 | 3.33  | 8.14  | 25.61 | 5.05  | 10.75 | 15.74 | 43.52 | 17.00  |
> >
> > [3] https://github.com/zkshan2002/RTO?tab=readme-ov-file
> >
> > [4] https://huggingface.co/RLHFlow/LLaMA3-SFT

---

> ### Author Response · Authors · 2025-11-21
> **Q8: need more analysis on why thinking improves performance on the benchmark.**
>
> **A8:** We thank you for this insightful suggestion. In the revised Section 3.2.1, we will expanded our analysis to explain why the thinking process leads to superior performance on our benchmark, particularly for unconventional tasks. Drawing on recent literature (e.g., ICML25 Paper: Thinking LLMs: General Instruction Following with Thought Generation[Wu et al., 2024]), we attribute these improvements to three key mechanisms facilitated by the thinking stage:
> 1. Enhanced Instruction Understanding and Planning (Beyond Logic) While "Chain-of-Thought" is traditionally associated with math and logic, recent findings suggest its utility is much broader. We argue that for unconventional instructions, the thinking process acts as a "planning workspace."
> - Standard LLMs produce the first token of a response with a fixed compute budget. By generating thoughts first, the model extends its compute budget to better "digest" the instruction before committing to a response. As observed by Wu et al. , internal thoughts allow models to plan the overall structure of a response even in creative tasks (e.g., defining characters or tone in creative writing). In our benchmark, this allows the model to explicitly interpret unconventional constraints and plan its output structure, rather than relying on immediate, reflexive token generation.
> 2. Self-Refinement through "Drafting and Evaluating" The thinking process enables a mechanism of internal trial-and-error which is absent in direct-response models.
> - Thinking models can function similarly to System 2 thinking by generating a "draft response" and then evaluating it within the thought block. Wu et al. demonstrate that even for non-reasoning tasks (like general knowledge questions), models trained to think will "evaluate their draft response" to ensure satisfaction before outputting the final answer. This self-evaluation step is crucial for unconventional instructions where the "correct" path isn't a standard pattern in the pre-training data. The model can catch potential deviations from the user's unique constraints during the thought phase.
> 3. Better Information Retrieval and Contextualization For tasks requiring domain-specific or niche knowledge (often present in unconventional benchmarks), the thinking phase serves as a retrieval step.
> - Qualitative analysis shows that thinking helps models "recall relevant information" and organize key points (e.g., recalling specific soil properties for a cactus before writing a guide). This prevents the model from "hallucinating" a generic answer. By grounding the response in retrieved facts during the thought process, the final output becomes more accurate and tailored to the specific prompt.
> Summary:In conclusion, the performance gain on our benchmark is not merely due to increased reasoning capability, but because the thinking process equips the model with the ability to understand complex intent , plan structural requirements, and self-correct drafts. This confirms that "thinking" is a general-purpose capability that benefits a wide variety of instruction-following tasks, not just mathematical reasoning.

---

> ### Author Response · Authors · 2025-11-21
> **Q9: Why are some categories harder than others?**
>
> **A9:** We appreciate this insightful question. To rigorously address why difficulty varies across categories, we calculated the average performance of all 22 evaluated models for each instruction type.
>
> | Rank | Instruction Category | Avg Score | Key Conflict|
> |------|----------------------|----------------|--------------------------------------|
> | 1 | Question Correction (QC) | 34.51 | Helpfulness Bias|
> | 2 | Counter-Conventional Formatting (CCF) | 44.82 | Readability Bias |
> | 3 | Instructional Induction (II) | 53.60 | Pattern Matching |
> | 4 | Deliberately Incorrect Answers (DIA) | 58.52 |Factuality Bias|
> | 5 | Intentional Textual Flaws (ITF) | 59.83 | Fluency Bias |
> | 6 | Mid-turn Instruction Modification (MIM) | 63.41 | Context Tracking |
> | 7 | Counterfactual Answering (CA) | 70.10 | Context Grounding|
> | 8 | Code without Comments (CC) | 77.85 | Stylistic Constraints|
>
> Our analysis reveals that difficulty is not random; it is strongly correlated with the strength of the "Cognitive Inertia" induced by Supervised Fine-Tuning (SFT) and RLHF. Furthermore, we observe that even for categories deemed "easier" on average, certain top-tier models exhibit unexpected failures due to rigid adherence to specific training norms.
>
> 1. Hierarchy of Difficulty: The Impact of Cognitive Inertia
> Based on the average scores calculated from Table 3, we categorize the eight instruction types into three difficulty tiers:
>
> Tier 1: The Hardest (Avg Score < 45) – Direct Conflict with Alignment Objectives
>
> - Question Correction (QC, Avg ~34.5): This is consistently the most difficult category. SFT and RLHF heavily incentivize models to be "helpful" and provide answers even to noisy inputs. Instructions requiring the model to critique the prompt rather than solve it contradict this core alignment. For example, Qwen3-235B-Instruct scores only 40.28 on QC (English), failing to overcome its "helpfulness" bias.
>
> - Counter-Conventional Formatting (CCF, Avg ~44.8): Models are trained to prioritize readability (e.g., using markdown, lists). Instructions like "strictly avoid bullet points" force the model to suppress its highest-probability generation paths.
>
> Tier 2: Moderate Difficulty (Avg Score 50–65) – Conflict with Factuality and Fluency
>
> - Categories like Intentional Textual Flaws (ITF) and Deliberately Incorrect Answers (DIA) require the model to generate content that is factually incorrect or grammatically flawed. This fights against the model's fundamental language modeling objective (predicting correct tokens) and safety alignment (minimizing hallucinations).
>
> Tier 3: The "Easiest" (Avg Score > 70) – Alignment with Existing Task Paradigms
>
> - Code without Comments (CC, Avg ~77.9) and Counterfactual Answering (CA, Avg ~70.1) are generally easier because they align with familiar task structures. CA resembles "context-based QA" (like RAG), and CC merely requires omitting a component (comments) rather than generating incorrect content.
>
> 2. Nuance: Why Top Models Fail Even on "Easy" Categories
> Crucially, our data shows that a category having a high average score does not imply it is always easy. We observe a counter-intuitive phenomenon where highly capable models fail on "easy" tasks because they are over-aligned to specific "best practices."
>
> The "Good Habits" Trap in Code Generation (CC): While "Code without Comments" has the highest average score, specific high-performing models struggle severely here due to rigid training on "industry best practices" (where comments are mandatory). The Qwen3-235B-A22B-Instruct, despite being a massive model, scores an incredibly low 25.58 on the Chinese version of "Code without Comments" (CC). Similarly, DeepSeek-V3.1 scores only 25.58 on CC (Chinese) and 37.98 on CC (English), significantly below the average.
> These low scores suggest these models have extremely strong "coding style" inertia. They prioritize the SFT convention (writing commented code) over the user's explicit constraint (no comments), illustrating that stronger capability can sometimes lead to stronger inertia.
>
> The "Factuality" Blockade in Deliberately Incorrect Answers (DIA): Some models are so rigorously aligned to be factually correct that they cannot "lie" even when instructed to. GPT-OSS-120B, a strong open-source model, scores only 32.62 on DIA (English) , whereas the smaller 03-mini scores 93.91. This indicates that for certain models, the safety/factuality guardrails are too rigid to allow for flexible instruction following in adversarial contexts.
>
> Conclusion: The difficulty of a category is determined by the tension between the instruction and the model's training priors. However, this relationship is non-linear: highly aligned models may paradoxically perform worse on simple tasks (like removing comments) because their "cognitive inertia" is more entrenched. This underscores the necessity of Inverse IFEval for detecting overfitting to training conventions.

---

> ### Author Response · Authors · 2025-11-24
>
> Hi, we sincerely thank you very much for these constructive comments and evaluation of our manuscript. We would like to kindly ask you to take a look at our responses and reevaluate our work based on our clarifications. Please let us know whether our response addresses your concerns or whether there is any further detail we can provide to help address these concerns.
>
> Thank you again for dedicating your time to reviewing our paper.

---

> ### Author Response · Authors · 2025-11-26
>
> Hi, I hope this message finds you well. We are writing to gently follow up regarding our rebuttal. We understand that this is a busy period, but if you could spare a moment to review our responses and share any updated evaluations, we would be truly grateful.
>
> Please let us know if any additional clarification or information from our side would be helpful. We sincerely appreciate your time and effort in reviewing our work.

---

### Official Review · Reviewer_wX3K · 2025-11-01

**Soundness:** 3
**Presentation:** 3
**Contribution:** 3
**Rating:** 6
**Confidence:** 3

**Summary:**

This paper introduces Inverse IFEval, a benchmark that probes whether LLMs can follow counterintuitive instructions rather than defaulting to habits learned during SFT. The authors motivate a new evaluation dimension called Counterintuitive Ability and organize 8 instruction types such as question correction, intentional textual flaws, code without comments, counter-conventional formatting, deliberately incorrect answers, instructional induction, mid-turn instruction modification, and counterfactual answering. The dataset contains 1,012 items in Chinese and English across 23 domains, built through an expert-seeded, human-in-the-loop pipeline with LLM generation and careful filtering.

Experimentally, the study evaluates many closed and open models and finds that an o3 model leads the results, that thinking variants outperform non-thinking ones, and that reduced thinking budget models trail their full budget counterparts. Fine-tuned models perform worse on this benchmark, consistent with its goal of testing out-of-distribution instruction following. The authors also analyze test-time compute and show a clear Best-of-N effect, where increasing N from 1 to 16 to 32 raises scores, and several models approach or pass 90 at N equals 32. Together, these results suggest the benchmark exposes failure modes not captured by prior instruction following tests while leaving room for gains with better post-training.

**Strengths:**

1. Counterintuitive instructions provide a strong stress test of IF capability, revealing habitual biases and mode locking that standard prompts miss. This perspective fills a gap in existing evals.

2. The benchmark is carefully curated and well-documented, with bilingual coverage and diverse task types that the community can readily reuse. Its transparent construction and release improve reproducibility.

3. The authors evaluate a broad range of models and add thoughtful analyses of thinking modes and test-time compute. The comparisons yield actionable insights on trade-offs and failure modes.

**Weaknesses:**

1. **Insufficient annotation detail.** The authors lack a detailed description of the annotation process. For example, what is the detailed background of the "experts," and are they affiliated with the authors' team? What is the specific inter-rater agreement rate? Were there multiple rounds of iteration, and what was the proportion of data modified in each round? Also, clarify whether a written guideline existed and, if so, summarize key rules and provide examples of borderline cases and resolution criteria. Adding these details would give us a clearer understanding of the data quality.

2. **Unexplained outliers in results.** Some results appear to be strong outliers, for example Qwen3-235B-A22B-Thinking on QC English at 6.53 and DeepSeek-V3.1 on DIA English at 0.00. An analysis that probes these anomalies and discusses possible causes would be helpful.

3. **Limited error analysis.** The error analysis is mostly a few appendix case studies. A more systematic treatment that categorizes error types, quantifies their frequencies, and relates them to model families and test-time settings would make the findings more actionable.

**Questions:**

See the above weaknesses.

---

> ### Author Response · Authors · 2025-11-21
> **W1: Insufficient annotation detail**
>
> Thanks for your comments. We have addressed each of your concerns carefully.
> We sincerely thank you for your thorough review of our paper and your valuable feedback regarding the insufficient description of our annotation process. We acknowledge that the original manuscript lacked clarity on these details. Below, we provide a detailed supplementary description of the annotation process, which will be included in the appendix of the paper and referenced in Section 2.
>
> ### 1. Background of "Expert" Annotators and Their Relationship to the Authors
>
> The term "experts" in the paper refers to the core annotators responsible for seed question construction, category definition, and final quality control. Specific details are as follows:
>
> - **Core Expert Team**:
>   Consisted of 6 full-time professionals specializing in NLP/LLM training and data engineering, with an average relevant work experience of approximately 4.5 years. Four of them hold Master’s or Ph.D. degrees in machine learning or natural language processing. Their responsibilities included:
>   - Refining the definitions of the eight types of adversarial instructions and designing positive/negative examples.
>   - Manually writing and iteratively refining seed questions.
>   - Designing scoring rubrics and arbitrating disputed samples.
>
> - **Cross-functional Reviewers**:
>   We additionally invited 7 senior professionals from product, R&D, and operations teams to conduct independent quality checks, introducing diverse usage perspectives. Their main task was to verify "question clarity" and "alignment with real-world application scenarios." All cross-functional reviewers had at least one year of experience in using LLMs or data annotation.
>
> Regarding affiliation, all annotators belong to the same institution as the authors, but most annotators are not co-authors of this paper. The authors were primarily involved in task definition and overall framework design. During the actual annotation process, judgments on sample eligibility and refinements to the rubrics were independently carried out by the data team. The authors only provided arbitration opinions on a small number of disputed samples to minimize potential bias.
>
> We have supplemented the above background information in the paper to clarify the composition and roles of the "experts."
>
> ### 2. Inter-Annotator Agreement (IAA)
>
> As noted by the reviewer, we quantitatively assessed inter-annotator agreement at two levels:
>
> 1. **Agreement on Seed Data**
>    - During the seed question construction phase, each candidate seed question was independently reviewed by three core experts, who provided a binary judgment (eligible/ineligible).
>    - Before filtering, we calculated Fleiss’ κ based on all candidate seed questions, resulting in κ ≈ 0.71, with a raw agreement rate of 86%.
>    - Only questions unanimously judged as "eligible" by all three reviewers were included in the final seed set, ensuring 100% agreement on inclusion in the evaluation set.
>
> 2. **Agreement on Scaled Generated Data**
>    - For candidate samples generated by models and automatically filtered, we adopted a "2+1" strategy: each sample was independently reviewed by two annotators (for category consistency, clarity, and rubric applicability), with disagreements resolved by a senior expert.
>    - To quantify overall agreement, we performed repeated annotations on approximately 15% (randomly sampled) of the final evaluation set, calculating agreement based on "sample usability" and "assigned category label." The overall Fleiss’ κ was approximately κ ≈ 0.63, with a raw agreement rate of 81%. Category-specific agreement ranged between 0.56–0.72, indicating reliable differentiation and consistency across the eight instruction types.

---

> > ### Author Response · Authors · 2025-11-21
> > **W1: Insufficient annotation detail**
> >
> > ### 3. Multi-Round Iteration and Sample Revision Ratio
> >
> > Our construction process involved multiple iterative stages. The revision ratios across rounds are summarized below:
> >
> > 1. **Round 1: Structure and Category Verification (applied to all candidate samples)**
> >    - **Objective**: Ensure question clarity, absence of logical contradictions, and alignment with one of the eight predefined adversarial instruction types.
> >    - **Process**: Each question underwent manual review for "category consistency" and "clarity." Questions with mismatched categories, ambiguous instructions, or multiple adversarial factors were rewritten or discarded.
> >    - **Outcome**: Approximately 47% of candidate samples were modified (including rewriting, merging, or deletion).
> >
> > 2. **Round 2: Scoring Rubric Calibration and Edge Case Handling**
> >    - **Objective**: Ensure the scoring standards were operable and discriminative across different annotators.
> >    - **Process**: We ran models on a small pilot evaluation set, collected model outputs and annotator scores, analyzed disputed samples, and refined rubrics accordingly (e.g., splitting "partially follows instructions but retains training inertia" into finer levels).
> >    - **Outcome**: Approximately 24% of questions had their scoring rubrics adjusted or received explanatory notes, with less than 5% requiring complete rewriting or deletion.
> >
> > 3. **Round 3: Final Inspection and Sampling Review**
> >    - **Objective**: Conduct a final quality check to identify any residual ambiguities.
> >    - **Process**: Approximately 10% of questions were randomly selected for review by another group of experts, with particular focus on previously disputed samples.
> >    - **Outcome**: Only 6% of sampled questions underwent minor wording adjustments (e.g., replacing ambiguous terms, adding contextual cues), while the rubric structure remained largely stable.
> >
> > This three-round iterative strategy—comprising intensive initial screening, focused re-inspection, and final sampling—ensured diversity while minimizing ambiguous or multi-interpretable questions.

---

> > > ### Author Response · Authors · 2025-11-21
> > > **W1: Insufficient annotation detail**
> > >
> > > ### 4. Written Annotation Guidelines, Key Rules, and Edge Cases
> > >
> > > We used a comprehensive set of written annotation guidelines to direct the work of experts and reviewers. The guidelines consisted of four parts: (1) General Principles, (2) Formal Definitions and Examples for the Eight Instruction Types, (3) Scoring Rubric Level Descriptions, and (4) Common Edge Cases and Decision Procedures. A summary is included in the final paper. Key contents include:
> > >
> > > **(1) General Annotation Principles**
> > > - **Focus on "instruction compliance" over output "quality"**:
> > >   In this benchmark, models are penalized for deliberately deviating from instructions and reverting to training conventions—even if the output appears more "professional" or "correct" in a traditional sense.
> > > - **Single primary category per question**:
> > >   If a question involves multiple adversarial factors, annotators determine the primary training convention being tested and assign it to the dominant category; secondary factors are recorded as metadata tags rather than primary evaluation dimensions.
> > > - **Ambiguous samples must be rewritten**:
> > >   If two annotators disagree on which training inertia a question primarily tests, the question is not included in the evaluation set and must be rewritten or split into more specific instructions.
> > >
> > > **(2) Key Distinguishing Rules for the Eight Instruction Types (Selected Examples)**
> > > The final paper includes operational definitions for each instruction type, such as:
> > > - **Question Correction**: The question contains an obvious error or incompleteness, and the instruction explicitly requires the model to "first identify/correct the question before answering."
> > > - **Intentional Textual Flaws**: The question includes deliberately introduced typos, garbled text, or colloquial abbreviations, and the instruction explicitly requires the model to "ignore/retain these flaws," testing whether the model resists the impulse to "auto-clean" the text.
> > > - **Code without Comments**: The instruction explicitly requires "no comments" or "avoid explanations," contrary to common code training conventions (detailed comments, readability).
> > > - **Counter-Conventional Formatting**: The instruction requires the model to respond in an unstructured or non-listed format (e.g., "Do not use bullet points; write only one continuous paragraph"), opposing the common training convention of structured responses.
> > > - **Deliberately Incorrect Answers**: The instruction explicitly requires "deliberately providing a wrong answer/misleading explanation."
> > > - **Instructional Induction**: The question is designed to resemble a "typical template-style problem," but the instruction requires the model to avoid using conventional solution patterns, testing its ability to suppress learned templates.
> > > - **Mid-turn Instruction Modification**: Instructions are altered or supplemented mid-dialogue, requiring the model to override previous response strategies.
> > > - **Counterfactual Answering**: The instruction requires the model to generate answers based on a world-setting contrary to known facts (e.g., "Assuming Earth had two moons, answer the following...").

---

> > > > ### Author Response · Authors · 2025-11-21
> > > > **W1: Insufficient annotation detail --- 4.3**
> > > >
> > > > **(3) Edge Cases and Decision Criteria Examples**
> > > > For edge cases, the guidelines provide typical examples and unified decision rules, such as:
> > > > - **Boundary 1: Question Correction vs. Intentional Textual Flaws**
> > > >   - *Example*: The question contains an obvious typo, and the instruction states, "Do not correct the typo; answer the question directly."
> > > >   - *Decision Rule*: If the main focus is testing whether the model resists auto-correction (i.e., complies with "do not correct"), classify as **Intentional Textual Flaws**; if the emphasis is on "first rewriting the question correctly before answering," classify as **Question Correction**.
> > > >
> > > > - **Boundary 2: Deliberately Incorrect Answers vs. Counterfactual Answering**
> > > >   - *Example A*: The instruction is "Please deliberately write an incorrect explanation of a physical law."
> > > >   - *Example B*: The instruction is "In a fictional world with zero gravity, explain the phenomenon of object falling."
> > > >   - *Decision Rule*: If the instruction emphasizes "incorrectness itself" without relying on a systematic world assumption, classify as **Deliberately Incorrect Answers**; if the question constructs a self-consistent counterfactual world and requires the model to respond within that premise, classify as **Counterfactual Answering**.
> > > >
> > > > - **Boundary 3: Code without Comments vs. Counter-Conventional Formatting**
> > > >   - *Example*: The instruction requires "Do not write comments, and do not use bullet points; write only one compact code paragraph."
> > > >   - *Decision Rule*: Annotators determine the primary focus: if the core is "violating code commenting conventions," classify as **Code without Comments**; if the core is "violating structured output conventions," classify as **Counter-Conventional Formatting**. If two annotators disagree, the question is split or rewritten so that each addresses only one primary dimension.
> > > >
> > > > For such edge cases, we employed a "dual independent annotation + expert arbitration" process: if two annotators disagreed on the primary category, a senior expert provided a written decision rationale, which was then incorporated into the annotation guidelines to ensure consistency in subsequent labeling.

---

> ### Author Response · Authors · 2025-11-21
> **W2: Unexplained outliers in results.**
>
> We sincerely appreciate your meticulous observation of the experimental results. The reported outlier of DeepSeek-V3.1 on Deliberately Incorrect Answer (DIA) English at 0.00 was a result of a typo error during the final data entry for the manuscript, which is not due to any methodological or code-related issues. The correct result should be 28.7.
>
> The result for Qwen3-235B-A22B-Thinking on Question Correction (QC) English at 6.53 is also an outlier, which we have re-verified and confirmed through re-testing. The results are largely consistent with the values in the paper. It is indeed significantly low. Our error analysis confirms that this model family indeed performs poorly on the QC task. The primary residual errors include:
>
> 1. Incompute correction and calculation error: The model fails to clearly point out the error in the question and simultaneously provides an incorrect calculated result.
>
> 2. Mismatched answer identification: The model explicitly states the correct answer but then provides the wrong final choice.
>
> 3. Knowledge Omission and Error: The model fails to explicitly state that all three choices are correct and provides an incorrect final answer.
>
> We will correct this erroneous value in the final manuscript and include a footnote or a dedicated paragraph in the "Experimental Results" section to explain these identified outliers and their underlying causes.

---

> ### Author Response · Authors · 2025-11-21
> **W3: Limited error analysis**
>
> We appreciate this constructive suggestion. We agree that a more systematic and in-depth error analysis is crucial for better diagnosing models' systematic weaknesses in counterintuitive instruction-following.
>
> We conducted an extensive error analysis on five model families (GPT, Claude, Gemini, Qwen, DeepSeek), focusing on their specific failure modes across eight inverse instruction task categories. If the paper is accepted, this detailed analysis will be integrated into the "Further Analysis" section of the manuscript.
>
> 1. BO1 analysis for five model families.
>
> We systematically annotated and statistically analyzed the BO1 inference results for five model families across eight inverse instruction tasks. Each error sample was labeled with a two-tier tagging system:
> - Inverse Instruction Category: (e.g., Question Correction, Intentional Textual Flaws, Code without Comments, Counter-Conventional Formatting, Deliberately Incorrect Answer, Instructional Induction, Mid-turn Instruction Modification, Counterfactual Answering).
> - Fine-Grained Failure Mode: (e.g., "still generated comments," "insisted on using structured bullet points," "answered correctly instead of deliberately incorrect," "refused to execute adversarial instructions," "ignored counterfactual setting, reverted to real-world knowledge," "overly complicated simple questions," "failed to identify the latest instruction in dialogue").
> Based on these labels, we categorized the observed errors into three major types, corresponding to underlying Training Inertia
>
> | Category          | Description                                                                       |
> |-------------------|-----------------------------------------------------------------------------------|
> | Formatting Inertia | Errors related to code commenting, text normalization, and structural formatting  |
> | Knowledge Inertia  | Errors related to deliberately incorrect answers and counterfactual scenarios     |
> | Logic Inertia      | Errors related to problem correction, instruction induction, and secondary instruction modification (Instruction-Structure) |
>
> We found that for all model families, 45%-59% of errors were concentrated in Formatting Inertia tasks. This highlights that adherence to "standardized output formatting" is the most entrenched type of training inertia.
>
> In contrast, GPT-series models showed a significantly higher proportion of errors in Knowledge Inertia tasks (around 31%, compared to 10%-23% for other families). This suggests that the instruction to violate knowledge (e.g., "answer incorrectly," "counterfactual settings") conflicts more sharply with the model's alignment strategies.
>
> 2. Verification with high decoding budgets (BO16 / BO32)
>
> To confirm the robustness of these findings, we further selected three representative models (gpt-oss-120b-0805-high, QwenAPI-3-235b-a22b-thinking, DeepSeekAPI-R1-0528) and performed a systematic error analysis on the remaining errors under stronger test-time settings (BO16 and BO32), using the same annotation system.
>
> We find the three training inertia categories (Formatting Inertia, Knowledge Inertia, Logic Inertia) remained the dominant error sources, showing a highly stable and distinct distribution across families.
>
>   1. GPT: Errors were primarily driven by Knowledge Inertia (up to $\approx 52\%$ in BO16), typically refusing to provide incorrect answers or reverting to factual knowledge under clear counterfactual settings.
>   2. Qwen: Errors were almost entirely dominated by Logic Inertia (up to $\approx 89\%$ in BO32), indicating it is highly susceptible to being "railroaded" by its standard problem-solving templates and parsing routines.
>   3. DeepSeek: Errors were influenced by a dual inertia of Formatting Inertia and Logic Inertia (Formatting Inertia $\approx 40\%$, Logic Inertia $\approx 60\%$), with Knowledge Inertia errors consistently below $2\%$.
>
>   We also provide task-granularity insights in BO16 and BO32 settings.
>
>   1. Deliberately Incorrect Answer: Residual errors in BO16 were almost exclusively contributed by GPT, while Qwen and DeepSeek showed very few such errors.
>
>   2. Code without Comments: DeepSeek exhibited a significantly higher error rate here, suggesting its prior preference for "commented, structured, standardized code style" is the most difficult to override.
>
>   3. Question Correction / Instructional Induction: Qwen’s residual errors were highly concentrated here, frequently forcing questions requiring simple correction or direct answers back into its complex, standard analytical templates learned during training.
>
>   Thees analysises strongly indicate that the failures in Inverse IFEval are not primarily due to a lack of basic capability, but rather the models' difficulty in overriding the deeply reinforced answering paradigms and alignment strategies acquired during training, aligning precisely with our design intent of probing "inverse cognitive capability."

---

> ### Author Response · Authors · 2025-11-24
>
> Hi, we sincerely thank you very much for these constructive comments and evaluation of our manuscript. We would like to kindly ask you to take a look at our responses and reevaluate our work based on our clarifications. Please let us know whether our response addresses your concerns or whether there is any further detail we can provide to help address these concerns.
>
> Thank you again for dedicating your time to reviewing our paper.

---

### Official Review · Reviewer_Q9R6 · 2025-11-04

**Soundness:** 3
**Presentation:** 3
**Contribution:** 3
**Rating:** 6
**Confidence:** 4

**Summary:**

This paper addresses the cognitive inertia of Large Language Models (LLMs)—their struggle to follow counterintuitive instructions conflicting with training conventions—by proposing Inverse IFEval, a novel benchmark. It introduces "Counterintuitive Ability" to measure LLMs’ capacity to override training-induced biases.
Inverse IFEval includes 8 challenging task types (e.g., Question Correction, Deliberately Incorrect Answers, Counter-Conventional Formatting) and a 1012-sample dataset , built via a human-in-the-loop pipeline (observation→seed construction→large-scale generation→automatic filtering→human verification). An optimized "LLM-as-a-Judge" framework, with task-specific judge models and refined prompts, achieves 98% evaluation accuracy.
Experiments on various LLMs (closed-source like o3-high, open-source like Qwen3) reveal: thinking models outperform non-thinking ones, larger models perform better, and fine-tuned models lack flexibility. The work fills gaps in LLM evaluation, serving as a diagnostic tool and guiding future alignment to enhance LLMs’ reliability in real-world unconventional scenarios.

**Strengths:**

The idea is neat and interesting. The paper’s emphasis on Large Language Models (LLMs)’ ability to comply with adversarial instructions that conflict with training inertia fills the gap in the existing LLM evaluation landscape regarding the "unconventional instruction-following" dimension. It further complements instruction-following benchmarks of the IFEval category, and to some extent, provides a `lower bound' for instruction-following capability assessment. Additionally, several experimental findings are particularly insightful: for instance, instruction-finetuned models exhibit a decline in long-tail instruction-following capability, which can be attributed to their overfitting to the paradigms of Supervised Fine-Tuning (SFT).

**Weaknesses:**

Even though the paper provides a critical extension to instruction following evaluation, there are some aspects that I think the paper can further improve on.

Firstly, the relevance between the benchmark’s tasks and real-world scenarios requires enhancement. Some counterintuitive instructions designed in the work are overly extreme and not aligned with actual user long-tail needs (e.g., deliberately generating texts with a fixed number of typos), which may lead to a discrepancy where models performing well on the benchmark still struggle to address practical unconventional requirements.

Then, the analysis of the root causes of models’ cognitive inertia is insufficient. The paper only observes surface phenomena—such as the underperformance of fine-tuned models and the superiority of thinking models—but fails to delve into how core factors (e.g., supervised fine-tuning data scale, reinforcement learning with human feedback (RLHF) feedback types, or model architectural characteristics) shape cognitive inertia, making it difficult to fundamentally explain the mechanisms behind the observed performance differences.

Thirdly, the error analysis could be improved with in-depth analysis. While the paper presents isolated error cases of several representative models (e.g., Claude-4-Opus, Doubao-1.6-Thinking), it does not summarize common error patterns across different instruction types or explore cross-linguistic error regularities, limiting the insights into models’ systematic weaknesses in counterintuitive instruction-following.

Additionally, if the pos training phase integrates counterintuitive instruction training into their development pipelines, would the current 8 task types in Inverse IFEval lose their ability to effectively discriminate model capabilities? Do you have any idea how to avoid this problem?

**Questions:**

NA

---

> ### Author Response · Authors · 2025-11-21
> **W1: Relevance between the benchmark’s tasks and real-world scenarios**
>
> Thanks for your careful reviewing and constructive suggestions. We have addressed all your concerns below point-by-point.
>
> **A1**: We appreciate your analysis and acknowledge your concern regarding the connection between our benchmark tasks and typical real-world applications. We agree that the instructions in our benchmark deliberately deviate from the most commonly used, conventional commands. The rationale for this design choice is detailed below.
>
> 1. Relevance to long-tail and niche real-world applications
> While our instructions are unconventional, many instruction types in our dataset do find applications in specific, niche, or long-tail real-world scenarios. For instance, the requirement to avoid bullet points or numbered lists is highly relevant in contexts like voice interface systems or generating continuous narrative summaries, where rigid formatting is undesirable for smooth output delivery. The instruction to "deliberately generate texts with a fixed number of typos" can be commonly found in educational settings for automatically generating error correction exercises or language assessment materials for students.
>
> 2. Ensuring robustness and preventing data memorization
> The nature of long-tail requirements is that they are inherently difficult to exhaustively enumerate, categorize, and are constantly evolving. For example, model trainers from earlier years would never have imagined that a question like "How many 'r's are there in 'Strawberry'?" would become a popular way for users to test a large model’s capabilities. Extreme instruction types are a more stable and reliable way to define the category of "long-tail problems." This ensures that the benchmark remains effective for an extended period in assessing a model's true instruction-following capability. Besides, by reducing the overlap with typical real-world commands, we prevent models from relying on learned patterns in the training data. This design allows us to isolate the model's true generalized understanding of a new instruction.
>
> 3. Relevance to practical unconventional requirements
> Our benchmark is fundamentally designed to test the model's ability to truly comprehend the instruction. To achieve this, we have deliberately reduced the complexity of the execution task itself (i.e., after the instruction is understood). Although the instructions are unconventional, the required execution steps are generally simple. Our hypothesis is that models performing well on our benchmark demonstrate superior capability in understanding complex, arbitrary, and unconventional constraints. We believe that this superior foundation in Instruction Comprehension will logically translate to a better performance in interpreting the instructions associated with practical unconventional requirements.

---

> ### Author Response · Authors · 2025-11-21
> **W2: Analysis of the root causes of models’ cognitive inertia**
>
> **A2**:
> We sincerely thank the reviewer for this excellent suggestion. We agree that our initial analysis of the root causes of models' cognitive inertia was insufficient. Acting on this valuable feedback, we have designed new experiments to investigate these core factors more deeply and will incorporate the findings into a new "Further Analysis" section of our paper. We believe this will significantly enhance the value of our work and its contribution to the open-source community.
>
> **1. SFT**
>
> To analyze the impact of supervised fine-tuning (SFT) data scale, we conducted an experiment by fine-tuning the Qwen3-14B-Base model on the full Infinity-Instruct chat dataset (660k samples). We evaluated the performance of seven checkpoints throughout a single training epoch. Given the training constraints, we consider the performance of the official Qwen3-14B-Instruct as a practical upper bound, as it was undoubtedly trained on a much larger and more diverse dataset using more advanced techniques.
>
> As shown in Table 1, our results reveal a clear trend. In the initial training phase (0-500 steps), the model's performance improved substantially, with the overall score on our benchmark rising from 8.99 to 18.28. However, in the subsequent phase (500-3000 steps), the performance gains plateaued, showing much slower growth. While increasing data exposure clearly boosts the model's general instruction-following capabilities, even the extensively trained Qwen3-14B-Instruct achieved a modest score of only 34.38 on our benchmark, despite the simplicity of our instructions. In stark contrast, it scores 85.4 on IFEval (strict prompt), a standard instruction-following benchmark.
>
> These findings strongly suggest that existing SFT datasets lack coverage of the specific instruction-following scenarios captured by our benchmark. This discrepancy not only highlights the unique value of our benchmark in identifying these blind spots but also points to insufficient data coverage as a key root cause of cognitive inertia. To mitigate this issue, future work will need to focus on curating SFT datasets with broader and more diverse coverage.
>
> **Table 1:**
> | Model | II | QC | ITF | CCF | CC | DIA | MIM | CA | Overall |
> | :--- | :--- | :--- | :--- | :--- | :--- | :--- | :--- | :--- | :--- |
> | Qwen3-14B-Base | 12.99 | 12.22 | 3.48 | 4.88 | 3.03 | 8.06 | 7.40 | 22.22 | 8.99 |
> | Qwen3-14B-SFT-500-Step | 30.52 | 17.78 | 12.79 | 29.27 | 3.53 | 10.22 | 19.44 | 37.04 | 18.28 |
> | Qwen3-14B-SFT-1000-Step | 25.97 | 31.11 | 16.28 | 31.71 | 4.55 | 9.68 | 25.93 | 37.96 | 20.16 |
> | Qwen3-14B-SFT-1500-Step | 29.22 | 14.44 | 8.14 | 29.27 | 8.59 | 8.06 | 22.22 | 41.67 | 18.77 |
> | Qwen3-14B-SFT-2000-Step | 36.36 | 23.33 | 20.93 | 39.02 | 4.54 | 6.99 | 24.07 | 47.22 | 22.33 |
> | Qwen3-14B-SFT-2500-Step | 37.01 | 22.22 | 8.14 | 34.15 | 6.06 | 9.14 | 20.37 | 32.41 | 19.56 |
> | Qwen3-14B-SFT-3000-Step | 40.26 | 26.67 | 8.14 | 39.02 | 9.60 | 11.29 | 18.52 | 38.89 | 22.43 |
> | Qwen3-14B-Instruct | 44.81 | 33.33 | 18.60 | 65.85 | 21.21 | 15.59 | 43.52 | 56.48 | 34.38 |
> | Qwen3-32B-Instruct | 64.07 | 32.59 | 24.81 | 36.70 | 51.22 | 59.14 | 51.85 | 63.58 | 49.28 |
>
> **2. RLHF**
>
> Regarding the analysis of “reinforcement learning with human feedback (RLHF) feedback types,” we designed a set of experiments using open-sourced models from “DPO Meets PPO: Reinforced Token Optimization for RLHF” [3]. We tested various models trained with different RLHF algorithms—including DPO, PPO, RTO, and SimPO—all built upon the same base model, Llama-3-8B-SFT [2]. As shown in Table 2, none of the RLHF algorithm variants led to improved performance on our benchmark. This suggests that many existing RLHF methods may not adequately account for the instruction scenarios represented in our benchmark. In some cases, the reward models used in these algorithms might even exacerbate the model’s cognitive inertia toward conventional instruction formats, ultimately resulting in performance degradation on our tasks.
>
> **Table 2:**
> | Model | II | QC | ITF | CCF | CC | DIA | MIM | CA | Overall |
> | :--- | :--- | :--- | :--- | :--- | :--- | :--- | :--- | :--- | :--- |
> | Llama-3-8B-SFT | 17.53 | 2.22 | 9.30 | 13.41 | 1.01 | 11.83 | 16.67 | 39.81 | 13.14 |
> | Llama-3-8B-DPO | 22.08 | 0.00 | 5.81 | 12.20 | 2.52 | 13.44 | 7.41 | 36.11 | 12.45 |
> | Llama-3-8B-PPO | 16.88 | 10.00 | 6.98 | 7.32 | 2.02 | 12.37 | 11.11 | 35.19 | 12.25 |
> | Llama-3-8B-SimPO | 13.64 | 8.89 | 6.98 | 12.20 | 1.51 | 6.99 | 12.96 | 23.15 | 9.88 |
> | Llama-3-8B-RTO | 9.09 | 10.00 | 1.17 | 6.10 | 1.51 | 11.83 | 9.26 | 28.70 | 9.39 |
> | Llama-3-8B-Instruct | 30.52 | 3.33 | 8.14 | 25.61 | 5.05 | 10.75 | 15.74 | 43.52 | 17.00 |
>
> [1] https://huggingface.co/datasets/BAAI/Infinity-Instruct
>
> [2] https://huggingface.co/RLHFlow/LLaMA3-SFT
>
> [3] https://github.com/zkshan2002/RTO?tab=readme-ov-file

---

> > ### Author Response · Authors · 2025-11-21
> > **W2: Analysis of the root causes of models’ cognitive inertia**
> >
> > **3. model architectural characteristics**
> >
> > Regarding the analysis of "model architectural characteristics," we have included the test results of Qwen3-32B-Instruct from the original paper in Table 1. It can be observed that a larger number of parameters generally endows the model with better capabilities, as it provides stronger representational capacity and learning potential, enabling the model to understand and execute complex instructions in a more profound and nuanced manner. However, this may also introduce greater "Knowledge Inertia" to the model. For instance, Qwen3-32B-Instruct performs significantly worse than Qwen3-14B-Instruct on the Counter-Conventional Formatting task. This suggests that future research should focus more on mitigating the tendency of models to merely learn to recognize and adhere to correct output formats, and instead enhance their ability to comprehend the semantic meaning of task instructions. This point is also discussed in the study "Do Models Really Learn to Follow Instructions? An Empirical Study of Instruction Tuning" [4].
> >
> > **Part of Table 1:**
> > | Model | II | QC | ITF | CCF | CC | DIA | MIM | CA | Overall |
> > | :--- | :--- | :--- | :--- | :--- | :--- | :--- | :--- | :--- | :--- |
> > | Qwen3-14B-Instruct | 44.81 | 33.33 | 18.60 | 65.85 | 21.21 | 15.59 | 43.52 | 56.48 | 34.38 |
> > | Qwen3-32B-Instruct | 64.07 | 32.59 | 24.81 | 36.70 | 51.22 | 59.14 | 51.85 | 63.58 | 49.28 |
> >
> > [4] https://aclanthology.org/2023.acl-short.113.pdf

---

> ### Author Response · Authors · 2025-11-21
> **W3: In-depth error analysis**
>
> **A3**: We appreciate this constructive suggestion. We agree that a more systematic and in-depth error analysis is crucial for better diagnosing models' systematic weaknesses in counterintuitive instruction-following.
> We conducted an extensive error analysis on five model families (GPT, Claude, Gemini, Qwen, DeepSeek), focusing on their specific failure modes across eight inverse instruction task categories. If the paper is accepted, this detailed analysis will be integrated into the "Further Analysis" section of the manuscript.
>
> **1. BO1 analysis for five model families.**
>
> We systematically annotated and statistically analyzed the BO1 inference results for five model families across eight inverse instruction tasks. Each error sample was labeled with a two-tier tagging system:
> - Inverse Instruction Category: (e.g., Question Correction, Intentional Textual Flaws, Code without Comments, Counter-Conventional Formatting, Deliberately Incorrect Answer, Instructional Induction, Mid-turn Instruction Modification, Counterfactual Answering).
> - Fine-Grained Failure Mode: (e.g., "still generated comments," "insisted on using structured bullet points," "answered correctly instead of deliberately incorrect," "refused to execute adversarial instructions," "ignored counterfactual setting, reverted to real-world knowledge," "overly complicated simple questions," "failed to identify the latest instruction in dialogue").
>
> Based on these labels, we categorized the observed errors into three major types, corresponding to underlying Training Inertia:
>
> | Category | Description | Proportion of total errors |
> | :--- | :--- | :--- |
> | Formatting Inertia | Errors related to code commenting, text normalization, and structural formatting | Approximately 50% |
> | Knowledge Inertia | Errors related to deliberately incorrect answers and counterfactual scenarios | Approximately 25% |
> | Logic Inertia | Errors related to problem correction, instruction induction, and secondary instruction modification (Instruction-Structure). | Approximately 25% |
>
> We found that for all model families, 45%-59% of errors were concentrated in Formatting Inertia tasks. This highlights that adherence to "standardized output formatting" is the most entrenched type of training inertia.
>
> In contrast, GPT-series models showed a significantly higher proportion of errors in Knowledge Inertia tasks (around 31%, compared to 10%-23% for other families). This suggests that the instruction to violate knowledge (e.g., "answer incorrectly," "counterfactual settings") conflicts more sharply with the model's alignment strategies.
>
> **2. Verification with high decoding budgets (BO16 / BO32)**
>
> To confirm the robustness of these findings, we further selected three representative models (gpt-oss-120b-0805-high, QwenAPI-3-235b-a22b-thinking, DeepSeekAPI-R1-0528) and performed a systematic error analysis on the remaining errors under stronger test-time settings (BO16 and BO32), using the same annotation system.
>
> We find the three training inertia categories (Formatting Inertia, Knowledge Inertia, Logic Inertia) remained the dominant error sources, showing a highly stable and distinct distribution across families.
>
>   1. GPT: Errors were primarily driven by Knowledge Inertia (up to $\approx 52\%$ in BO16), typically refusing to provide incorrect answers or reverting to factual knowledge under clear counterfactual settings.
>
>   2. Qwen: Errors were almost entirely dominated by Logic Inertia (up to $\approx 89\%$ in BO32), indicating it is highly susceptible to being "railroaded" by its standard problem-solving templates and parsing routines.
>
>   3. DeepSeek: Errors were influenced by a dual inertia of Formatting Inertia and Logic Inertia (Formatting Inertia $\approx 40\%$, Logic Inertia $\approx 60\%$), with Knowledge Inertia errors consistently below $2\%$.
>
>   We also provide task-granularity insights in BO16 and BO32 settings.
>
>   1. Deliberately Incorrect Answer: Residual errors in BO16 were almost exclusively contributed by GPT, while Qwen and DeepSeek showed very few such errors.
>
>   2. Code without Comments: DeepSeek exhibited a significantly higher error rate here, suggesting its prior preference for "commented, structured, standardized code style" is the most difficult to override.
>
>   3. Question Correction / Instructional Induction: Qwen’s residual errors were highly concentrated here, frequently forcing questions requiring simple correction or direct answers back into its complex, standard analytical templates learned during training.
>
>   These analysis strongly indicates that the failures in Inverse IFEval are not primarily due to a lack of basic capability, but rather the models' difficulty in overriding the deeply reinforced answering paradigms and alignment strategies acquired during training, aligning precisely with our design intent of probing "inverse cognitive capability."

---

> ### Author Response · Authors · 2025-11-21
> **W4: Addressing future training and benchmark validity**
>
> **A4**: We acknowledge this potential concern, as data contamination is a serious threat to the longevity of any diagnostic benchmark. We have discussed this internally and concluded that the issue can be effectively mitigated through several protective factors inherent in our design:
>
> 1. Lack of standardized ground truth
>
> Our benchmark does not provide conventional ground truth answers for instruction tuning. Instead, we only provide a checklist-based evaluation protocol detailing the required constraints. This unique setup creates a significant obstacle for directly incorporating our data into standard instruction-tuning pipelines, as trainers cannot easily define the desired output for a general instruction set.
>
> 2. The unconventional nature as a protective barrier
>
> The unconventional and often inverse nature of our instructions serves as an inherent protective barrier:
> - Trade-off Risk: Utilizing our data as general training data for instruction tuning poses a significant risk. Forcing a model to adhere to inverse instructions (e.g., repeatedly generating Counterfactual Answers or Deliberately Incorrect Answers) may negatively impact its performance on other conventional, high-value benchmarks, as it directly conflicts with established alignment goals (truthfulness, helpfulness, coherence).
>
> - Deterrent Effect: The inherent unconventionality of the instructions acts as a deterrent, as optimizing for our benchmark often requires violating typical training norms.
>
> 3. Rapid iteration and unreleased test sets
>
> We maintain a high level of agility in our data management through the following strategies:
>
> - Data Synthesis Pipeline: We possess an efficient data synthesis pipeline that enables us to rapidly iterate on new versions of our instruction set.
>
> - Test Set Rotation: We will maintain a dedicated set of unreleased test data that is independent of the public dataset. This allows us to continuously assess models using novel, uncontaminated data, thus preserving the benchmark's validity over time.
>
> - Limited Scale: Even if the public data were used for training, the relatively small scale (an average of $126.5$ instructions per category) means it would be challenging for models to achieve strong generalization across all eight inverse instruction types.
>
> In conclusion, the combination of non-standard labeling, task unconventionality, and our ability to quickly generate independent, high-quality test data should effectively address the threat of data contamination. If our paper is accepted, a thorough discussion of this limitation and our mitigation strategy will be added to the "Limitation" section.

---

> ### Author Response · Authors · 2025-11-24
>
> Hi, we sincerely thank you very much for these constructive comments and evaluation of our manuscript. We would like to kindly ask you to take a look at our responses and reevaluate our work based on our clarifications. Please let us know whether our response addresses your concerns or whether there is any further detail we can provide to help address these concerns.
>
> Thank you again for dedicating your time to reviewing our paper.

---

### Author Response · Authors · 2025-11-29
**Summary of Major Revisions and Clarifications**

Thanks for handling/reviewing our submitted manuscript: "**Inverse IFEval: Can LLMs Unlearn Stubborn Training Conventions to Follow Real Instructions?**". We would like to thank the reviewers for their insightful and constructive comments and suggestions. By addressing each of the issues raised by reviewers, we believe that the quality and clarity of our paper can be improved a lot. The major responses are summarized as follows:

1. **Real-world Relevance & Design Rationale:** We have extensively discussed the alignment between our designed instruction types and real-world scenarios, providing concrete examples. Furthermore, we have elaborated on the motivation behind our instruction design and provided a multi-faceted justification for its rationality. (See Reviewer Q9R6.W1, Reviewer mHLJ.Q1, Q2 & Q3)

2. **Root Cause Analysis of Cognitive Inertia:** We designed and conducted supplementary experiments to analyze the root causes of models’ "cognitive inertia" across four key dimensions: Data, SFT, RLHF, and Model Architecture. The results further highlight the value and necessity of the instruction types designed in our benchmark. (See Reviewer Q9R6.W2, Reviewer mHLJ.Q7)

3. **Systematic Error Analysis:** We performed a systematic error analysis by classifying error types and calculating their frequencies. We also analyzed these errors in correlation with specific model families and test-time settings to provide deeper insights. (See Reviewer Q9R6.W3, Reviewer wX3K.W3)

4. **Data Contamination & Robustness:** We investigated potential data leakage risks, demonstrated the robustness of our benchmark against such leakage, and proposed specific countermeasures to mitigate these risks. (See Reviewer Q9R6.W4)

5. **Annotation Details:** We have added comprehensive details regarding the annotation process to ensure readers have a clear understanding of the data quality and rigorousness. (See Reviewer wX3K.W1)

6. **Reliability of Evaluation:** We have provided supplementary arguments and evidence to demonstrate the reliability and validity of our evaluation methodologies. (See Reviewer mHLJ.Q5 & Q6)

7. **Outlier Analysis:** We verified, rectified, and provided a detailed analysis of the significant outliers observed in the experimental results. (See Reviewer wX3K.W2)

8. **Safety Risk Assessment:** We analyzed potential safety risks associated with the method and proposed a viable solution to address these concerns. (See Reviewer mHLJ.Q4)

9. **Reasoning Models in OOD Tasks:** We explored the reasons behind the superior performance of reasoning models (e.g., CoT-based models) on Out-Of-Distribution (OOD) instruction-following tasks. (See Reviewer mHLJ.Q8)

10. **Difficulty Variance Analysis:** We statistically analyzed the difficulty differences across various instruction types within our benchmark and investigated the underlying causes. (See Reviewer mHLJ.Q9)

Again, we would like to sincerely thank you very much for these constructive comments and evaluation for our manuscript.

---

### Meta-Review · Area_Chair_tqYf · 2026-01-07

**Summary:**

The paper introduces Inverse IFEval, a benchmark designed to evaluate Large Language Models (LLMs) on their ability to follow counterintuitive instructions that conflict with training-induced conventions. By formalizing the notion of Counterintuitive Ability and curating a dataset of 1,012 bilingual samples across eight task types, the authors aim to fill a gap in current evaluation frameworks that largely overlook unconventional or out-of-distribution (OOD) instruction-following.

**Reviewer Concerns:**

- Real-world relevance: Some benchmark tasks (e.g., deliberately inserting typos) feel contrived and may not reflect genuine user needs. The connection between benchmark performance and practical utility requires stronger justification.
- Limited causal analysis: While the paper observes performance differences across model types, it does not sufficiently probe the underlying mechanisms (e.g., data scale, RLHF feedback, architectural choices) that drive cognitive inertia.
- Annotation transparency: Details of the annotation process are sparse. Information on annotator expertise, inter-rater agreement, iteration rounds, and guidelines would strengthen confidence in dataset quality.
- Error analysis: Current treatment is cursory, relying on isolated case studies. A systematic categorization of error types, their frequencies, and cross-model comparisons would yield deeper insights.

**Reviewer Scores:**

Reviewer mHLJ may have a chance to raise the score.

---

### Decision · Program_Chairs · 2026-01-26

Accept (Poster)